# Distilled Pretraining: A modern lens of Data, In-Context Learning and Test-Time Scaling

**Sachin Goyal**[*]
FAIR at Meta
Carnegie Mellon University
sach.goyalsachin@gmail.com

**David Lopez-Paz**
FAIR at Meta
dlp@meta.com

**Kartik Ahuja**
FAIR at Meta
kartikahuja@meta.com

## Abstract

In the past year, distillation has seen a renewed prominence in large language model (LLM) *pretraining*, exemplified by the Llama-3.2 and Gemma model families. While distillation has historically been shown to improve statistical modeling, its effects on new paradigms key to modern LLMs—such as *test-time scaling* and *in-context learning*—remain underexplored. In this work, we make three main contributions. First, we show that pretraining with distillation yields models that exhibit remarkably better test-time scaling. Second, we observe that this benefit comes with a trade-off: distillation impairs in-context learning capabilities, particularly the one modeled via induction heads. Third, to demystify these findings, we study distilled pretraining in a sandbox of a bigram model, which helps us isolate the common principal factor behind our observations. Finally, using these insights, we shed light on various design choices for pretraining that should help practitioners going forward.

## 1 Introduction

Knowledge distillation, first proposed by Buciluă et al. (2006) for compressing ensembles, was later popularized by seminal works of Ba & Caruana (2014) and Hinton et al. (2015). However, distillation didn't trickle into the pipelines of early large language models (LLMs)—such as GPT-2/3 and Llama 1/2. But more recently, distillation has resurged as a prominent method in the LLM landscape, not just during post-training, but also *pretraining* as seen in the Llama-3.2 (Meta AI, 2024b) and Gemma (Gemma et al., 2024; 2025). This shift reflects a growing reality: extremely large models (e.g., Llama-4-Behemoth (Meta AI, 2024a)) are too costly to deploy widely and will increasingly serve solely as teachers for distilling smaller, more practical models. Going forward, these deployed models are likely to be pretrained entirely via distillation as seen in Llama-4-Maverick (Meta AI, 2024a) that was distilled from Llama-4-Behemoth.

Despite its growing role, the science of distillation (using soft labels) in modern LLM *pretraining* has remained largely unexplored. Gemma-3 and Llama-3.2 models show clear empirical benefits on standard benchmarks from pretraining with distillation. However, these models typically leverage teachers trained on far more data than the students. This raises a fundamental question: are the gains from distillation merely a result of additional teacher data, or do they reflect unique benefits beyond extra data exposure? As we hit the data wall, will distillation continue to be beneficial? Moreover, modern LLMs are no longer limited to evaluation on standard benchmarks. *New paradigms such as in-context learning and test-time scaling are key to current LLM frontiers, yet the effect of pretraining with distillation on these paradigms remains largely unexamined.*

In this work, we uncover key trade-offs associated with distilled pretraining (DPT). First, we show that DPT remains beneficial on standard language modeling tasks, even in the data-constrained regime where the student and the teacher models are trained on the same data. This suggests promise for scaling DPT further. However, in contrast, we observe that *naively scaling pretraining with distillation (DPT) hurts the in-context learning performance* (Figure 1b). In particular, distillation impairs the learning of induction heads (Olsson et al., 2022)—the transformer circuits that enable models to search and copy from context(Figure 1c).

---

[*]Work done at Meta

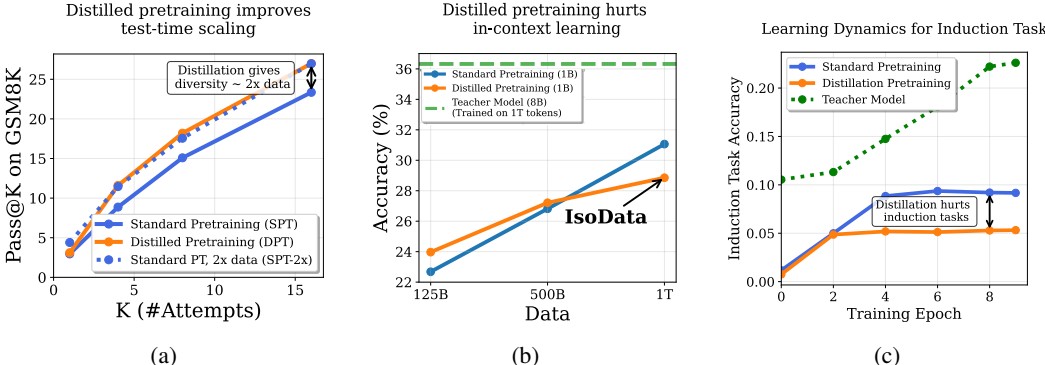

Figure 1: **Distilled pretraining in modern LLM regime (a)** Comparing standard pretraining (SPT) with distilled pretraining (DPT). On reasoning tasks like GSM8k, although both the models have a similar pass@1, DPT substantially outperforms SPT on pass@$k$ for higher $k$ (27% vs 23% for $k = 16$). Infact, DPT matches the pass@16 of a standard pretrained model trained on twice the data. **(b)** Distilled pretraining hurts in-context learning capabilities when the student and teacher model see the same data. In the figure, as we scale the student data to 1T (data seen by the teacher), the gains of distillation over standard pretraining on in-context learning tasks diminish (Figure 3 for details). **(c)** We demystify these findings by analyzing a bigram sandbox, where we show that training with distillation can impair the learning of induction heads (Bietti et al., 2023), which form the key mechanism behind in-context learning.

Strikingly, the very process of distillation that undermines in-context learning, at the same time also yields models that demonstrate *markedly better test-time scaling capabilities*. We study this through pass@$k$, where the model is allowed multiple attempts per question. Distilled models outperform standard pretraining on pass@$k$ at larger $k$, even when pass@1 is the same (Fig.1a). On GSM8k, for example, both models have the same pass@1, but the distilled model achieves a much higher pass@16—27% versus 23%. Remarkably, it even matches the pass@16 of a standard-pretrained model trained on twice the data, despite a lower pass@1. Similar patterns hold on MATH and MBPP, where distilled pretraining consistently improves test-time scaling by enhancing generation diversity(Dang et al., 2025).

Interestingly, the mechanisms through which distillation undermines in-context learning are the same ones that enhance test-time scaling. We study this tradeoff in a simple yet expressive sandbox of a bigram model (Bietti et al., 2023; Edelman et al., 2024). A bigram model is characterized by a matrix in which each row represents the next token probability distribution over the vocabulary. Pretraining with distillation is beneficial in learning the high-entropy rows. These rows basically model prompts like "I work at", which admit multiple valid completions (e.g., "gym", "hospital", "restaurant"). In contrast, distillation does not help in learning low-entropy rows which model the deterministic state transitions (prompts), e.g., induction heads where the next-token probability distribution is one-hot. For these cases, distillation does not provide any information beyond what is already there in ground truth one-hot labels. Worse, an imperfect teacher can hurt the learning of these low-entropy rows by introducing noise via soft probability distribution(Figure 1c).

Finally, borrowing insights from our analysis, we discuss various design choices for improving pretraining in §5. These include *distillation-specific data curation*, teacher selection, and comparisons with other recent advances such as multi-token prediction (Gloeckle et al., 2024), which we hope will aid practitioners going forward. We summarize our key contributions in this work below:

- **Test-time scaling:** We show that distilled pretraining produces models with markedly stronger test-time scaling, often matching standard pretraining on up to twice the data.

- **In-context learning trade-off:** We find that these gains come at a cost, as distillation impairs in-context learning, particularly by weakening induction heads.

- **Bigram analysis:** We isolate the common mechanism that drives the improvements in test-time scaling but impairs in-context learning at the same time.

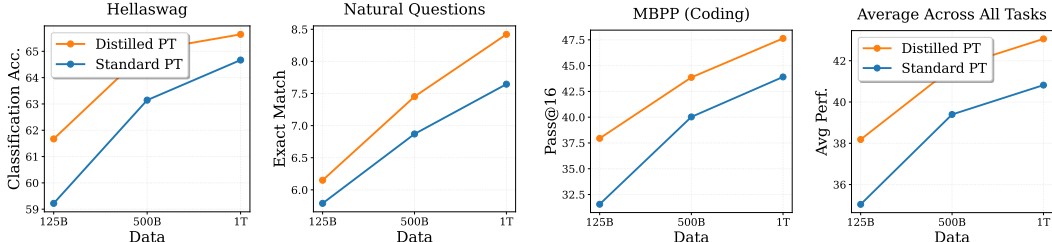

Figure 2: **IsoData Distillation (§ 2)**: Will distilled pretraining remain effective when the student and teacher are trained on the same data? To explore this, we use an 8B model trained on 1T tokens as a teacher. Using this teacher, we train various student models, with and without distillation, scaling up the data to the exact same 1T tokens. We observe that even in the IsoData case where both teacher and student have seen the same 1T tokens, the distilled model generally outperforms standard pretraining on standard language modeling tasks. Thus distillation generally remains beneficial even in a data-constrained regime. See Figure 12 for more tasks.

- **Practitioner Takeaways:** We translate these insights into concrete design choices for improving pretraining with distillation, including distillation specific data curation, teacher selection, etc.

## 1.1 PRELIMINARIES

We recall the distillation with soft label objective (Hinton et al., 2015), which interpolates between fitting hard labels $y_i$ and teacher soft labels $s_i = \sigma(h_{\text{teacher}}(x_i)/T)$:

$$h^\dagger \in \arg\min_{h \in \mathcal{H}} \frac{1}{n}\Big[(1-\alpha)\sum_{i=1}^{n}\ell(y_i, \sigma(h(x_i))) + \alpha\sum_{i=1}^{n}\ell(s_i, \sigma(h(x_i)))\Big]. \quad (1)$$

In LLM pretraining, the same objective applies to next-token prediction with $y_i$ replaced by $x_{j+1}$. Full derivations and notation are deferred to the Appendix B.

## 2 NO EXTRA DATA: DOES DISTILLATION STILL IMPROVE PERFORMANCE?

Recent pretrained LLM families—such as the Gemma-3 and Llama-3.2 series—have shown clear benefits from distillation compared to training from scratch. However, these models typically leverage teachers trained on significantly more data than the students ultimately use, raising a fundamental question: Are the gains from distillation simply due to this additional teacher data? Does distillation offer unique benefits beyond merely seeing extra data via the teacher in the modern web-data regime?

We begin this work by answering the basic question raised above via a set of "IsoData Distillation" experiments. We first train an 8B teacher model on 1T tokens. We then train 1B students—with and without distillation—on the same 1T tokens to test if distillation still helps when both see identical data. Figures 2 and 12 compare the performance of the two 1B models on standard language modeling tasks like COPA, HellaSwag, NaturalQA, TQA, GSM8k, etc. We observe that distillation continues to benefit even when training is scaled to the same data as the teacher (1T tokens).

> Distilled pretraining (DPT) continues to be generally beneficial even in the data-constrained regime, when the student is shown the same amount of data as the teacher.

Theoretically, Mobahi et al. (2020); Nagarajan et al. (2024) study self-distillation and understand why it improves performance, which is kind of a IsoData distillation. We refer the reader to Appendix H for a detailed discussion on theoretical works in IsoData distillation. In the next section, we will analyze distilled pretraining on new paradigms centric to modern LLMs, beyond the standard language modeling tasks: in-context learning and test-time scaling.

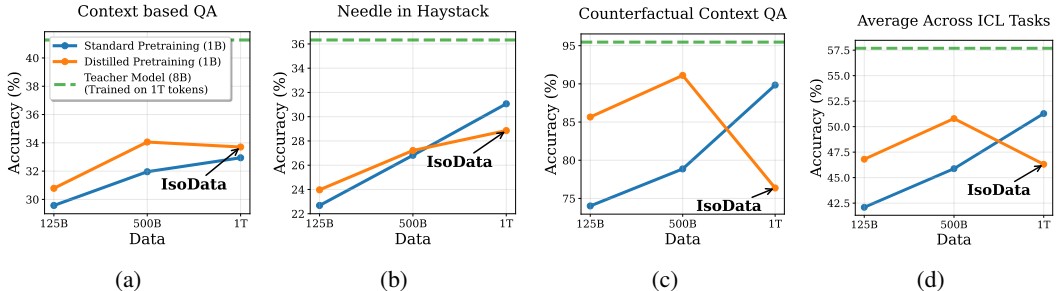

Figure 3: **Distilled pretraining impairs in-context learning, especially in the IsoData setting (§ 3.1):** We train 1B models with and without distillation using a 8B teacher trained on 1T tokens. We observe that the advantages of distillation on in-context learning tasks diminish as the amount of training data increases (each scatter is a separate model trained with a full LR scheduler). Eventually, the distilled model underperforms in the IsoData setup, where both the teacher and student are trained on the same data. This is because induction heads which form a key mechanism behind in-context learning (Olsson et al., 2022) are built on low-entropy mappings, requiring the model to copy a specific token from earlier in the sequence. For these cases, distillation can't help—it can only match the hard label at best, and at worst, it actively hinders learning for such copying tasks by softening the supervision. This is in contrast to performance on standard language modeling tasks where distillation continues to help even in IsoData setting (Figure 2).

## 3    Distilled Pretraining Through the Modern Lens: In-Context Learning and Test-Time Scaling

Knowledge distillation has long been shown to improve *in-weights learning* (IWL), resulting in stronger performance in standard evaluation tasks and benchmarks (Gemma et al., 2024; 2025). However, in modern LLMs, the desired capabilities extend much beyond the classical setting of IWL. The ability to generate *diverse solution paths* is critical for skills like test-time scaling and search at inference (Chow et al., 2024; Dang et al., 2025; Chen et al., 2025), but more crucially, to also enable better post-training with reinforcement learning with verifiable rewards (RLVR). Likewise, *in-context learning* (ICL)—where models learn and adapt from inference time prompts is especially desirable. In this section, we examine how pretraining with distillation shapes these two capabilities key to the current LLM frontiers: test-time scaling and in-context learning (ICL).

### 3.1    Distillation impairs in-context learning

The seminal work of Olsson et al. (2022) introduced induction heads as a key mechanism behind in-context learning in modern LLMs. They enable models to "copy" tokens from earlier positions in the input into later parts of the output (Olsson et al., 2022; Edelman et al., 2024; Bietti et al., 2023). For instance, given a prompt like "I work at Gym," an induction head helps the model replicate "Gym" in a follow-up question about workplace. This copying ability is critical for tasks requiring models to attend to and reuse information from the context.

**Experimental Setup:**    We train an 8B teacher on 1T tokens and 1B students—with and without distillation—on the same 1T tokens. We call this the "IsoData" setup, which ensures a fair comparison by removing any indirect data advantage for the distilled model. To measure model's ability to copy from context which is a hallmark for induction head learning— we use 3 benchmarks: (a) context-based QA (DROP (Dua et al., 2019), RACE (Lai et al., 2017)) (b) needle-in-a-haystack tasks (babilong (Kuratov et al., 2024)); and (c) counterfactual context QA  (Goyal et al., 2025), where the correct answer as per the context contradicts factual knowledge (i.e. answer based on model's memory or weights), forcing the model to rely solely on contextual cues. Conterfactual evaluations thus give a clearer signal regarding model's abilities to follow the input context (see samples in Appendix J.1).

**Observations:**    Figure 3 compares the in-context learning performance of the two 1B models trained with and without distillation, as the training data is scaled to 1T tokens. We observe a consistent

pattern that as the training tokens are increased, the relative advantage of distillation over the standard pretrained model keeps on diminishing. Infact the distilled model eventually underperforms in the IsoData setup (1T tokens) on needle-in-haystack and counterfactual-QA tasks. These observations are in stark contrast to the observations on standard language modeling tasks (e.g., Hellaswag, GSM8k, NaturalQA) in Figure 2, where distillation continues to offer advantage even in the "IsoData" setup.

For counterfactual context-based QA, models often default to using their parametric knowledge which is incorrect. This tendency grows with scale or when confidence in the context-based answer is low. As a result, distilled models show a accuracy drop compared to the smaller-scale (500B tokens) model. The distilled model is expected to outperform standard pretraining in the non-IsoData setting (e.g., 125B and 500B training in Figure 3) due to the teacher's indirect data advantage.

> **Why does distillation hurt in-context learning (ICL)?** ICL is driven by induction heads that implement low-entropy copy mappings, where the model must reproduce a token from earlier in the sequence. For such deterministic targets, distillation adds no signal—a perfect teacher's soft labels collapse to the one-hot ground truth, so DPT can only match hard supervision. In practice, imperfect teachers assign non-zero mass to distractors, softening supervision and injecting noise into an otherwise clean mapping, which can hinder learning of the copy circuit. As a result, DPT often fails to help—and can hurt—induction-head formation and in-context learning. We formalize this effect in our bigram sandbox in § 4.

## 3.2 DISTILLATION HELPS DIVERSITY

**Experimental Setup:** We train 1B models on 125B tokens, with and without distillation, using the Llama-3.1-8B base as teacher. We later also consider an IsoData setting to isolate the effect of extra data the teacher may have seen. For distilled pretraining (DPT), we study two settings: DPT-50 and DPT-90, where the distillation loss is weighted at 50% and 90% respectively ($\alpha$ in Eq. 2). We compare models **under two settings**: (1) using a sampling temperature that maximizes pass@16, and (2) sweeping temperature from 0 to 1.5 (in increments of 0.1) and plotting pass@1 vs. pass@16. This lets us distinguish whether a model is simply stronger overall (higher pass@1 and pass@16), or whether it has higher generation diversity—achieving better pass@16 despite similar pass@1. We clarify that in this work we focus on generation diversity as measured by pass@$k$, a standard metric in the LLM reasoning literature (Chen et al., 2025; Dang et al., 2025; Chow et al., 2024).

**Distilled pretraining unlocks superior test-time scaling.** In Figure 4 (top row), we first compare the pass@$k$ curves for standard pretraining (SPT model) and distilled pretraining (DPT-50 model with 50% weight of distillation). We begin by selecting the sampling temperature that maximizes pass@16 performance (a full temperature sweep analysis follows next). Observe in Figure 4(a,b) that while the DPT-50 model has slighlty worse pass@1 compared to the SPT model, the DPT-50 model obtains a much higher pass@16 (e.g., 28% vs. 23% on GSM). Infact on MATH (Figure 4b), the DPT-50 model even starts off worse than SPT on pass@$k$ at $k = 1$, but clearly outperforms it as $k$ increases—exhibiting a *striking crossover phenomenon*.

**Distilled pretraining gives diversity worth seeing $2\times$ data.** We now evaluate DPT against a harder baseline of standard pretraining on 2x data (SPT-2x , 250B tokens), increasing the distillation weight to 90% (DPT-90 ). As shown in Figure 4 (top row), DPT-90 achieves higher pass@16 than SPT-2x across all three benchmarks—even though it is trained on half the data and has a lower pass@1. This highlights the strong diversity gains in generations from distillation.

In Figure 4 (bottom row), we plot pass@1 vs. pass@16 across temperatures from 0 to 1.5. Across all benchmarks—GSM8k, MATH, and MBPP—the DPT-90 curve consistently lies vertically above the SPT-2x curve. That is, for any fixed pass@1, the distilled model achieves a higher pass@16. Note that both the models have the same maximum pass@1 (if one optimizes the temperature for pass@1), but the distilled model always has a higher maximum pass@16, or infact a higher pass@16 for any reasonable pass@1. This reinforces that distilled pretraining enables stronger test-time scaling.

**Diversity gains even in IsoData setting** For the results in Figure 4, we use Llama-3.1-8B as the teacher, trained on more data than our 1B students. Importantly, the gains persist in the IsoData setting

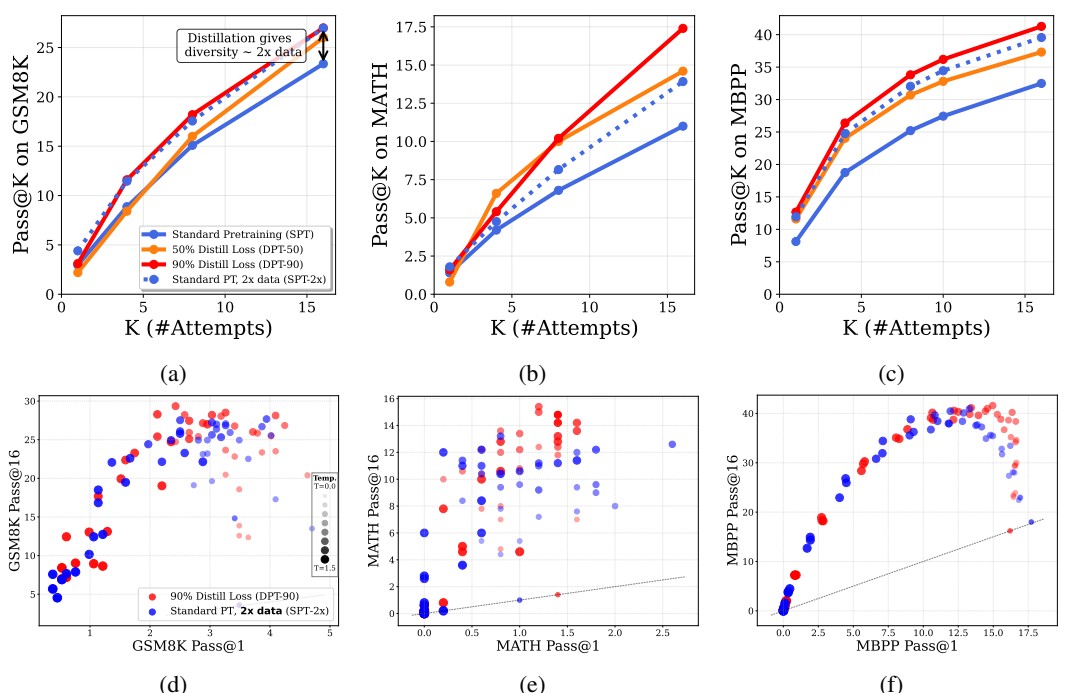

Figure 4: **Distilled pretraining improves generation diversity and enables superior test-time scaling(§ 3.2): Top-row (a-c):** We plot pass@$k$ curves with temperature first optimized for pass@16 performance. Distilled pretraining with 50% weight of distillation (DPT-50) consistently outperforms standard pretraining (SPT) on pass@16, even though it has worse pass@1 on GSM8k and MATH. **Bottom-row (d–f):** We increase the distillation weight to 90% (DPT-90) and compare against a stronger baseline of standard pretraining on *2x data* (SPT-2x). We plot pass@1 vs. pass@16 across temperatures 0–1.5 (step 0.1). DPT-90 consistently achieves higher pass@16 for any reasonable pass@1, despite using half the data, and attains the top pass@16 on all three benchmarks. This shows that distilled pretraining yields models with greater generation diversity and stronger test-time scaling.

where both teacher and student are trained on the same 1T tokens (Figure 12): distilled pretraining (orange curve) still outperforms standard pretraining on GSM8k and MBPP Pass@16. This shows that DPT's test-time scaling advantage cannot be explained solely by the teacher's access to more data.

**Why does distillation help with diversity?** When prompts admit multiple plausible continuations—like "I work at"—the ground truth data provides only one answer (e.g., hospital), but a teacher model distributes probability mass across many valid completions (e.g., hospital, gym, cafe). Distillation exposes the student to this richer signal, which intuitively explains why it improves the model's diversity in it's generations at inference time. We discuss this more formally in the next section.

> While pass@1 demands only that the top prediction be correct, pass@$k$ evaluates whether any of the $k$ outputs are valid—rewarding breadth over precision. This subtle shift means that correctly ranking one option is not enough; the model must distribute probability mass across multiple plausible answers. Distilled models excel at this, helping them exhibit better test-time scaling. We will discuss this in more detail in Appendix F.

## 4 BUILDING INTUITION VIA A BIGRAM SANDBOX

In the previous section, we saw that distilled pretraining improves test-time scaling but hurts in-context learning performance by hurting the learning of induction heads. In this section, we try to dissect the reasons behind this using a simple yet powerful sandox of a bigram model.

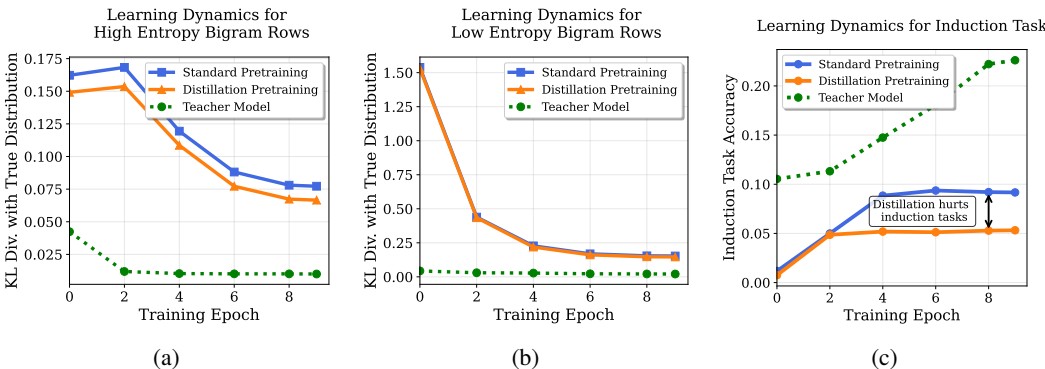

Figure 5: **Understanding distillation through the lens of a bigram model (§ 4):** To dissect why distillation enhances diversity yet impairs in-context learning, we examine these phenomena in a simple yet expressive sandbox—a bigram model (Bietti et al., 2023; Edelman et al., 2024). A bigram models a first-order Markov chain represented via a transition probability matrix. **(a)** We illustrate that distillation particularly aids the learning of high-entropy rows, corresponding to prompts such as "I work at", which admit multiple plausible completions (e.g., "gym", "hospital", "restaurant"). **(b, c)** Conversely, distillation offers no advantage for learning low-entropy rows (b), which not only represent deterministic state transitions (prompts), but are also essential for induction head formation as described by Bietti et al. (2023). Moreover, distillation with an imperfect teacher may even slow or hinder learning these low-entropy, induction-head-like rows (c).

## 4.1 BIGRAM MODEL: LOW-ENTROPY VS. HIGH-ENTROPY ROWS

To build intuition for our results, consider two illustrative prompts: (i) **Low Entropy Prompts:** "$2 + 3 =$" with completions: a) 5, b) 4, c) 7 — where a) occurs with probability 1 in natural data; and (ii) **High Entropy Prompts:** "I go to" with completions: a) office, b) gym, c) restaurant, d) 33 — where a), b), and c) each occur with probability $1/3$ in natural data.

**Bigram data generation process:** A bigram model captures a first-order Markov process, where the next token depends only on the current token. Mathematically, it is represented by a matrix $\pi \in \mathbb{R}^{k \times k}$, where each element $\pi_{ij}$ denotes the transition probability from token $i$ to token $j$. Our dataset consists of sequences generated from the above bigram model, and the first token is sampled uniformly from the vocabulary. We categorize each row of the transition matrix $\pi$ as either *low-entropy* or *high-entropy*, based on the entropy of that row relative to a fixed threshold. High-entropy rows are akin to prompts that have a diverse completion set (recall "I go to" example from above). Low entropy rows then correspond to prompts with less-diverse completions (e.g., "2+3=").

**Distillation accelerates learning of high-entropy bigram rows** In Figure 5(a,b), we present the results of the experiments. The teacher is a bigger model trained on 2x more data than the students (details in the Appendix E). We observe that models trained from scratch and models trained via distillation are both at par when it comes to the low entropy rows (Figure 5b). A real distinction appears in how well they approximate the high entropy rows, where the distilled model performs better, i.e., it requires fewer samples to achieve a better approximation of the high-entropy row(Figure 5a). We now formalize the intuitions behind the above arguments.

**Sample complexity analysis for bigram model** Each row of the bigram matrix $\pi$ is $p$-sparse, i.e., contains at most $p$ non-zero entries. We consider sequences of length two. Both the scratch-trained and distilled student models are parameterized by bigram matrices $\pi^{\text{scratch}}$ and $\pi^{\text{distill}}$, respectively, while the teacher is parameterized by $\pi^{\text{teacher}}$.

**Proposition 1.** *(informal) In bigram learning with p-sparse rows*
- *Sample complexity when training with distillation is $\mathcal{S}_{distill} = \mathcal{O}(k \log k)$.*
- *Sample complexity when learning bigram without distillation is $\mathcal{S}_{standard} \approx \frac{p}{\epsilon^2} \mathcal{S}_{distill}$, where $\epsilon$ is the upper bound on the the approximation error.*

We refer the reader to Appendix D for the proof. Consider first the high-entropy setting where the row sparsity $p = \mathcal{O}(k)$, where $k$ denotes the vocab size. The standard pretrained model requires $\mathcal{O}(k^2 \log k)$ samples, whereas the distilled model needs only $\mathcal{O}(k \log k)$. In contrast, in the low-entropy setting where $p$ is constant, both models have sample complexity at most $\mathcal{O}(k \log k)$. This reflects the empirical observations from Figure 5(a,b) where we observe distillation accelerating the learning of high-entropy rows but no difference for low-entropy rows.

## 4.2 WHY DOES INDUCTION HEAD LEARNING SLOW DOWN FOR DISTILLED MODELS?

Recall from § 3.1 that distillation impairs the learning of induction heads—key circuits for in-context learning. We revisit this phenomenon by detailing the induction head setup in our bigram sandbox. Following Bietti et al. (2023), we modify the bigram model to embed an *induction-style pattern* using **trigger tokens**. A trigger token is a special token such that whenever it appears, it is always followed by a fixed token within that sequence. This fixed token differs across sequences but remains the same for all trigger occurrences within a given sequence.

Formally, before generating each sequence, we randomly choose a "copy target" token $c \in \{1, \ldots, k\}$. We then alter the bigram transition matrix $\pi$ so that whenever the current token is the trigger (denoted $i = t$), the next token is deterministically $c$. Mathematically:

$$\tilde{\pi}_{ji} = \begin{cases} \pi_{ji} & \text{if } i \neq t \\ \mathsf{I}(j = c) & \text{if } i = t \end{cases}$$

Sampling from $\tilde{\pi}$ produces a setting where the optimal strategy is to learn to *copy* the token ($c$) following a trigger token (the token $t$ in the above case)—mimicking the behavior of induction heads in real LLMs (Olsson et al., 2022; Bietti et al., 2023). The difference between standard pretraining and distillation emerges in the supervision signal.

- In standard pretraining, encountering a trigger yields a one-hot ground-truth label for the next token—clean and unambiguous supervision.
- In distillation with a *perfect* teacher, the soft label distribution is also exactly one-hot, so the supervision is identical. In practice, however, teachers are imperfect: they may assign non-zero probability mass to distractor tokens. This produces a slightly higher-entropy target distribution, effectively injecting noise into what should be a deterministic mapping.

## 4.3 WHY DOES pass@$k$ IMPROVE FOR DISTILLED MODELS?

We earlier observed in Figure 4 that distilled models can have a higher pass@k despite having a lower or a similar pass@1 compared to a standard pretrained model. In Appendix F, we show that optimal pass@k performance requires accurate estimation of the underlying probability distributions rather than just correct ordering of class probabilities (which suffices for pass@1). Distilled models, by incorporating soft supervision from teachers, better approximate these probability distributions, especially in high-entropy settings where multiple valid answers exist. We study this theoretically in the setting of estimating a Bayes optimal classifier in Appendix F.

## 5 TOKEN ROUTING: MITIGATING THE DROP IN IN-CONTEXT LEARNING

In Section 3.1, we saw that distilled models underperform on ICL tasks because they are based on low-entropy mappings where distillation doesn't help. To mitigate this, we propose a simple yet effective strategy: token routing. Recall from Equation 2 that during distilled pretraining, there are two terms in the loss-one for loss with ground truth labels and the other with teacher's label

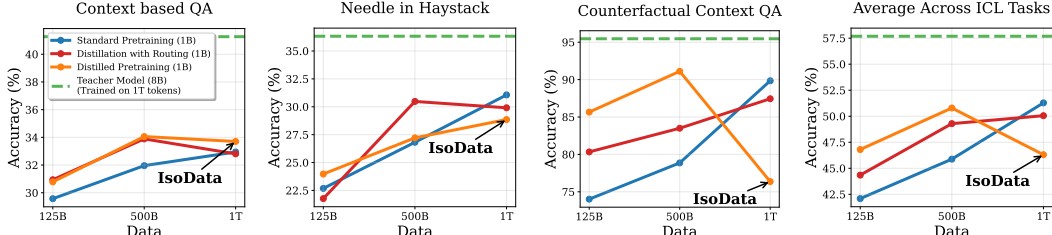

Figure 6: **Token Routing: Mitigating the Drop in In-Context Learning(§ 5):** Distilled models often struggle on ICL tasks due to softening of supervision on low-entropy (near-deterministic) tokens. To mitigate this, we apply token routing: for each input, we skip the distillation loss on the 15% lowest-entropy tokens, using only ground-truth supervision there. This strategy (red curve) improves over vanilla distillation (orange). As shown in Table 1, these gains come without hurting standard language modeling performance.

(distillation loss term). Rather than applying distillation loss with the teacher's label on all tokens, we dynamically adjust the supervision based on the entropy of the teacher's output. Specifically, given an input sequence, we first compute the teacher's soft labels for the sequence. We then drop the distillation loss term for $x\%$ of the positions with lowest entropy in teacher's label—falling back to only the standard hard-label supervision with the ground truth here.

In Figure 6, we show results with $x = 15\%$ token routing. On two of three tasks—Needle in a Haystack and Counterfactual QA—routing yields clear gains over vanilla DPT, partially closing the gap with standard pretraining. On Context-based QA, no gain is expected since vanilla DPT already outperforms standard pretraining. Importantly, routing low entropy tokens to standard pretraining objective does not hurt standard LM tasks (Table 1), reinforcing the view that distillation gains arise from high-entropy teacher labels. Appendix G.1 reports results with $x = 30\%$ routing, which offers no further ICL benefit and degrades standard benchmarks.

**Practitioners guidelines:** While preliminary, token-routing demonstrates how distillation centric data curation pipelines can help and we hope our work motivates future research in this direction. Beyond token routing, we explore several additional design choices crucial for effective distilled pretraining in Appendix G. These include comparing distillation against other diversity-enhancing methods like multi-token prediction, investigating the choice of teacher model (base vs. instruction-tuned vs. RL-trained), and analyzing the impact of top-k sampling during distillation. These important practical considerations that can significantly impact the effectiveness of distilled pretraining.

## 6 RELATED WORKS

**Modern paradigm of distillation** In the past year, we've witnessed a resurgence of distillation in the context of modern LLMs. Both the Llama-3.2 (1B and 3B models) (Meta AI, 2024b) and Gemma model families (sizes ranging from 3B to 27B) (Gemma et al., 2024; 2025) rely heavily on pretraining distillation mechanisms. These models primarily employ the prominent weighted loss introduced by Hinton et al. (2015). Both Llama-3.2 and Gemma series of models use teacher models that have been trained on way more data than the what the student model is ultimately trained on. In this work, we first show that distilled pretraining continues to help even in the IsoData setting. More crucially, later we isolate intriguing tradeoffs of distilled pretraining.

Synthetic data, generated by teacher models, is now commonly used to enrich pretraining corpora, effectively constituting another form of hard-label distillation. Cha & Cho (2025) analyzed distillation using synthetic data generation (hard-label distillation), where students learn from samples drawn directly from the teacher model. In contrast, our analysis focuses on Hinton et al. (2015)-style pretraining distillation, where students learn from soft labels provided by the teacher, leading us to distinct conclusions from Cha & Cho (2025) regarding prediction diversity. We believe that this happens because of the difficulty in sampling diverse synthetic pretraining data (hard labels) from the teacher. Recently, Busbridge et al. (2025) discuss how distillation might not be helpful under certain

compute-matched settings. However, in § 2 we argue that incorporating teacher logit computation cost might not be the correct setting, and it is more important to consider data-constrained settings.

**Improving distillation for LLMs**  Li et al. (2021) discuss using small teacher models for tokens where the student model predictions are less confident. Cho & Hariharan (2019); Zhang et al. (2024a); Mirzadeh et al. (2019); Zhang et al. (2023); Beyer et al. (2022) highlight bigger teacher is not always better and propose various ways to mitigate capacity mismatch between student and the teacher. Our practitioner's guidelines in this work are complementary to the above findings.

We refer the reader to Appendix H for additional related works and Appendix I for future directions.

## 7    REPRODUCIBILITY STATEMENT

To ensure the reproducibility of our findings, we provide comprehensive implementation details and experimental specifications throughout the paper and appendices. First, the full details of our training equation we used is explained in detail in Appendix B. Our experimental setup, including model architectures, hyperparameters, training procedures, and evaluation protocols are detailed in the "experimental setup" subsection in §3.1 and §3.2. The dataset composition and other hyper-parameters are detailed in Appendix C. All theorems and proofs are detailed in the relevant sections in the Appendix.

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

## A    LLM USAGE.

We used a large language model (LLM) as a general-purpose tool to assist with polishing the writing style. The LLM was not involved in research ideation, experiment design, or analysis. All technical content, results, and conclusions are entirely our own, and we take full responsibility for the final manuscript.

## B    PRELIMINARIES

We start by revisiting the setup of distillation from Hinton et al. (2015). We are given a dataset $\{(x_i, y_i)\}_{i=1}^n$ of inputs $x_i \in \mathbb{R}^d$'s and the labels $y_i's \in \Delta^{k-1}$, where $k$ is the number of classes and $\Delta^{k-1}$ is a probability simplex over those classes. Let us begin with the objective of training a model from scratch on the above data using cross-entropy loss $\ell$, $h^\star \in \arg\min_{h \in \mathcal{H}} \frac{1}{n} \sum_{i=1}^n \ell(y_i, \sigma(h(x_i)))$, where $h$ is a candidate function drawn from the hypothesis class $\mathcal{H}$, $\sigma : \mathbb{R}^k \to \Delta^{k-1}$ is the softmax function $\sigma_j(z) = \frac{\exp(z_j)}{\sum_{i=1}^k \exp(z_i)}$ and $\ell(y, \hat{y}) = -\sum_{j=1}^k y_j \log(\hat{y}_j)$.

We are now ready to define the standard objective used in distillation:

$$h^\dagger \in \arg\min_{h \in \mathcal{H}} \frac{1}{n} \Big[ (1-\alpha) \sum_{i=1}^n \ell(y_i, \sigma(h(x_i))) + \alpha \sum_{i=1}^n \ell(s_i, \sigma(h(x_i))) \Big], \tag{2}$$

where $\alpha \in [0, 1]$ and $s_i = \sigma(h_{\text{teacher}}(x_i)/T)$ is a soft label generated by the teacher using a temperature $T$.

This form of distillation has recently been adopted in pretraining language models as well. We first start by describing the next-token prediction objective over a sequence $(x_1, \cdots, x_t)$

$$\frac{1}{t} \sum_{j=1}^t \ell\Big(x_{j+1}, \sigma(h(x_{\leq j}))\Big) \tag{3}$$

The objective function used in pretraining distillation is

$$\frac{1}{t} \Big[ \sum_{j=1}^{t-1} (1-\alpha)\ell\Big(x_{j+1}, \sigma(h(x_{\leq j}))\Big) + \alpha \sum_{j=1}^{t-1} \ell\Big(s_{j+1}, \sigma(h(x_{\leq j}))\Big) \Big], \tag{4}$$

where $s_{j+1} = \sigma(h_{\text{teacher}}(x_{\leq j})/T)$

## C    GENERAL EXPERIMENTAL DETAILS

**Pretraining dataset composition**    Our pretraining corpus consists of tokens drawn from diverse domains to ensure broad coverage of knowledge and reasoning capabilities. The majority of the data comes from the DCLM (Li et al., 2025) like baseline dataset and GitHub repositories (neogithub, 2022). In addition, we include a range of specialized sources spanning mathematics, coding, scientific literature, and high-quality web content. Specifically, our mixture includes DeepMind Mathematics problems (Saxton et al., 2019), Proof Pile 2 collections (ArXiv, Open Web Math, Algebraic Stack) from Azerbayev et al. (2023), Stack Exchange from the pile (Gao et al., 2020), FineWeb-Edu (Lozhkov et al., 2024), and smaller curated sets such as Natural Reasoning Dataset (Yuan et al., 2025) and AQuA (Ling et al., 2017).

**Pretraining Hyperparameters**    For temperature(T) in distilled pretraining, we do a grid-search over $T \in \{0.5, 1, 2, 3\}$. We select the temperature which gives the best performance on standard benchmarks. In our experiments, $T = 1$ worked the best. We pretrain with cosine scheduler using a learning rate of $3e^{-3}$ for 1B models and $3e^{-4}$ for 8B models.

# D   PROPOSITION 1 (FORMAL).

**Proposition 1.**

- *If the number of sequences observed grow as $\mathcal{O}(k \log k + k \log(\frac{1}{\delta}))$, then the $\pi^{\text{distill}} = \pi^{\text{teacher}}$ with probability at least $1 - \delta$.*

- *If the number of sequences observed grow as $\mathcal{O}\left(\frac{(k \log k + (p/\epsilon^2 - 1)k \log \log k)}{\delta}\right)$, then for each row $i \in [k]$, $\mathbb{E}[\|\pi_i^{\text{scratch}} - \pi_i\|_1] \leq \epsilon$ with probability $1 - \delta$, where $\mathbb{E}$ is computed over the entire draw of the dataset.*

*Proof.* To prove the first part, let us recollect a standard result.

The coupon collector problem studies the following question. Suppose each box contains a coupon, and there $k$ different types of coupons. What is the number of boxes we need to see $T$ before we have collected all $k$ coupons? Assuming each coupon is drawn uniformly at random,

$$P(T > \beta k \log k) < k^{-\beta + 1}$$

Substitute $\beta = 1 + \frac{\log \frac{1}{\delta}}{\log k}$, we obtain

$$P(T > k \log k + k \log(\frac{1}{\delta})) < \delta$$

Translated to our setting, this means if we observe $k \log k + k \log(\frac{1}{\delta})$, then with probability at least $1 - \delta$ each of the distinct $k$ tokens have been observed at the first position in the sequence. This completes the proof for the first part.

We now turn to the model trained from scratch. The log-likelihood of a model is written as $\sum_{ij} n_{ij} \log(\hat{\pi}_{ij})$, where $n_j$ is the number of times we see a token $j$ appear after token $i$. The solution to maximum likelihood is simply $\hat{\pi}_{ij} = \frac{n_{ij}}{n_i}$, where $n_i = \sum_{j \in [k]} n_{ij}$. $\hat{\pi}_{ij}$ is an unbiased estimator of $\pi_{ij}$. Define

For this model, we need to ensure that each row in the estimated matrix is close to the true row. Next, we want to bound the distance between $\|\hat{\pi}_{i,:} - \pi_{i,:}\|_1$, where we particularly use $\ell_1$ distance to emphasize the role of sparsity. Observe that the variance of each element of the row is $\mathbb{E}[(\hat{\pi}_{ij} - \pi_{ij})^2] = \frac{\pi_{ij}(1 - \pi_{ij})}{\sum_j n_{ij}}$.

Observe that

$$\left(\mathbb{E}[|\hat{\pi}_{ij} - \pi_{ij}|]\right)^2 \leq \mathbb{E}[(\hat{\pi}_{ij} - \pi_{ij})^2] = \frac{\pi_{ij}(1 - \pi_{ij})}{\sum n_{ij}} \implies \mathbb{E}[|\hat{\pi}_{ij} - \pi_{ij}|] \leq \sqrt{\frac{\pi_{ij}(1 - \pi_{ij})}{\sum n_{ij}}} \quad (5)$$

To compute, $\|\hat{\pi}_i - \pi_i\|_1$, we only need to sum over the terms that are non-zero owing to the sparsity assumption. Suppose that without loss of generality first $p$ terms are non-zero. Hence, we obtain

$$\mathbb{E}[\|\hat{\pi}_i - \pi_i\|_1] = \sum_{j \leq p} \mathbb{E}[|\hat{\pi}_{ij} - \pi_{ij}|] \leq \sum_{j \leq p} \sqrt{\frac{\pi_{ij}(1 - \pi_{ij})}{\sum n_{ij}}} \quad (6)$$

We can arrive at a simple upper bound for $\sum_{j \leq p} \sqrt{\pi_{ij}(1 - \pi_{ij})}$ as follows. We again apply Cauchy-Schwarz inequality. We express

$$\sum_{j \leq p} \sqrt{\pi_{ij}(1 - \pi_{ij})} = \langle 1, [\sqrt{\pi_{i1}(1 - \pi_{i1})}, \sqrt{\pi_{i2}(1 - \pi_{i2})}, \cdots, \sqrt{\pi_{ip}(1 - \pi_{ip})}]\rangle$$

$$\langle 1, [\sqrt{\pi_{i1}(1 - \pi_{i1})}, \sqrt{\pi_{i2}(1 - \pi_{i2})}, \cdots, \sqrt{\pi_{ip}(1 - \pi_{ip})}]\rangle \leq \sqrt{p}\sqrt{\sum_j (\pi_{ij})(1 - \pi_{ij})} \leq \sqrt{p}$$

We substitute this in equation 6 to obtain

$$\mathbb{E}[\|\hat{\pi}_i - \pi_i\|_1] \leq \sum_{j \leq p} \sqrt{\frac{\pi_{ij}(1 - \pi_{ij})}{n_i}} \leq \sqrt{\frac{p}{n_i}} \qquad (7)$$

From the above, we can observe that if $n_i = \frac{p}{\epsilon^2}$, then

$$\mathbb{E}[\|\hat{\pi}_i - \pi_i\|_1] \leq \epsilon, \forall i \in [k]$$

Hence, if each token $i$ is observed at the first position of the sequence at least $\frac{p}{\epsilon^2}$, then we should obtain the desired outcome we set out to prove in this part.

We now recollect the generalized version of coupon collector's problem. In the generalized version one is interested in computing the number of boxes to collect defined as $T_m$ before collecting $m$ copies of each coupon. In this case,

$$\mathbb{E}[T_m] \approx k \log k + (m - 1)k \log \log k$$

If we apply Markov inequality on the above, we obtain a simple bound

$$P\left(T_m \geq \frac{1}{\delta} \cdot \mathbb{E}[T_m]\right) \leq \delta$$

Thus from the above, we gather that if the number of boxes collected is at least $\frac{1}{\delta} \cdot \left(k \log k + (m - 1)k \log \log k\right)$, then with probability at least $1 - \delta$ we have collected $m$ copies of each coupon.

We can now substitute $m = \frac{p}{\epsilon^2}$ to obtain our bound of $\frac{k \log k + (p/\epsilon^2 - 1)k \log \log k}{\delta}$. This completes the proof. $\qquad \square$

# E  EXPERIMENTAL DETAILS FOR BIGRAM SANDBOX AND INDUCTION HEAD LEARNING

Our bigram sandbox experiments were designed to provide a simple, controlled testbed for understanding how distillation influences test-time scaling and in-context learning. All results in Section 4 are derived from this setup.

**Data generation.**    The vocabulary consists of $k = 64$ tokens. The bigram transition matrix $\pi \in \mathbb{R}^{k \times k}$ was constructed to include a mix of low-, medium-, and high-entropy rows: low-entropy rows concentrated probability mass on 3–5 tokens; high-entropy rows were nearly uniform; medium-entropy rows had an intermediate profile. Trigger tokens were randomly selected (5, 10, or 20 triggers per experiment), with trigger-output mappings varying across sequences to induce induction head learning (following Bietti et al. (2023)). Sequences were generated using a first-order Markov chain with these bigram transitions, with special logic to ensure copying behavior for trigger tokens.

**Models.**    Both teacher and student models were implemented as small Transformers with 2–4 layers, causal masking, and a fixed sequence length of 64. Teacher models used 128-dimensional embeddings; students used 64-dimensional embeddings. Training was performed with Adam optimizer and a cosine learning rate schedule.

**Training.**    Teacher models were trained on datasets of size 16k sequences. Student models were trained with either cross-entropy (CE) loss or knowledge distillation (KD), using soft logits from the teacher. Dataset sizes for students were 8k sequences i.e. half the data. The KD objective used temperature $T = 2.0$ and mixing coefficient $\alpha = 0.5$ (Equation 2).

**Evaluation.**  All models were evaluated on a fixed held-out dataset of 4k sequences. Metrics included: Induction head accuracy (trigger → copy) as shown in Figure 1c; and KL-divergence between the ground-truth distribution (bigram rows) and the learnt distribution for low-, medium-, and high-entropy rows as shown in Figure 5.

Our full codebase will be released for reproducibility.

## F   WHY DOES pass@$k$ IMPROVE FOR DISTILLED MODELS?

**Demistifying** pass@$k$ **trends:**  In Figure 1(b), we saw a puzzling finding. The distilled model can start with a worse pass@1 and can have a much better pass@$k$. Is this a mere accident, or does there exist a deeper principle behind the observations?

Suppose that our data consists of one fixed prompt $x$, which is followed by three options $y = \{0, 1, 2\}$. The true probabilities are $p(y = 0|x) = \frac{1}{2} + \epsilon$, $p(y = 1|x) = \frac{1}{2} - \epsilon$ and 0 with $\epsilon > 0$. Define three classifiers:

- **Bayes optimal classifier, C1:** Assigns a probability 1 to class 0 and achieves the optimal pass@1 accuracy of $\frac{1}{2} + \epsilon$.

- **Diverse classifier with right coverage, C2:** Assigns a probability of $\frac{1}{2}$ to both classes 0 and 1. This classifier achieves a suboptimal pass@1 accuracy of $\frac{1}{2}$.

- **Diverse classifier with wrong coverage, C3:** Assigns a probability of $\frac{1}{2}$ to classes 0 and 2. This classifier achieves a suboptimal pass@1 accuracy of $\frac{1}{4} + \frac{\epsilon}{2}$.

Interestingly, observe that the pass@$k$ accuracy of **C1** is $\frac{1}{2} + \epsilon$ for all $k$. The pass@$k$ accuracy of **C2** is $1 - (\frac{1}{2})^k$ for all $k$. The pass@$k$ accuracy of **C3** is $(\frac{1}{2} + \epsilon)(1 - (\frac{1}{2})^k)$ for all $k$. As shown in Figure 7, the classifier **C2** exhibits crossover over the Bayes optimal classifier **C1**. Thus, the Bayes optimal classifier is suboptimal at higher pass@$k$. Further, **C3**'s support does not contain the support of the true distribution, highlighting the importance of right coverage over the correct solution space.

The above example leaves us with the question that if the Bayes optimal classifier is not optimal for pass@$k$, then what is? We derive this classifier below.

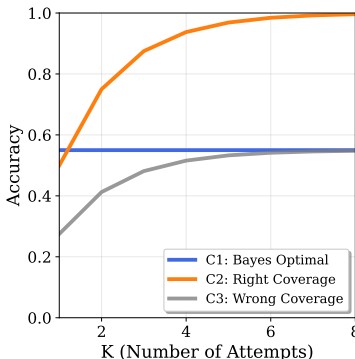

Figure 7: **Bayes optimal for** pass@1 **is not optimal for** pass@$k$**.** A diverse classifier with correct coverage (C2) outperforms the Bayes optimal classifier (C1) at higher $k$, while incorrect coverage (C3) remains suboptimal. Coverage—not just pass@1—is key to improving pass@$k$.

**Generalized Bayes optimality for** pass@$k$   In this section, we restrict ourselves to binary classification tasks with the true probability distribution over labels $y$ conditional on $x$ denoted as $p(y|x)$.

Recall the definition of a Bayes optimal classifier for binary classification. For each $x$ in the support of the training distribution, the classification rule is

$$\begin{cases} 0 & p(y = 0|x) > \frac{1}{2} \\ 1 & p(y = 1|x) \leq \frac{1}{2}. \end{cases} \tag{8}$$

Define a general classifier which assigns a probability $\alpha(x)$ to class 1 and $\beta(x)$ to class 0.

**Theorem 1.** *The generalized Bayes optimal classifier that achieves the optimal* pass@k *assigns for each $x$ in the training distribution*

$$\alpha^*(x) = \frac{\left(\frac{p(y=1|x)}{p(y=0|x)}\right)^{\frac{1}{k-1}}}{1 + \left(\frac{p(y=1|x)}{p(y=0|x)}\right)^{\frac{1}{k-1}}}. \tag{9}$$

*Proof.* pass@k accuracy of a classifier checks if at least one of the $k$ attempts of the classifier predicts the label correctly. For a fixed $x$ in the support of the training distribution, the pass@k accuracy of this classifier is stated as

$$p(y=1|x)\big(1 - (\beta(x))^k\big) + p(y=0|x)(1 - (\alpha(x))^k). \tag{10}$$

To understand the above expression, let us look at the first term. Conditional on $y = 1, x$, $\big(1 - (\beta(x))^k\big)$ is the probability that at least one of the attempts by the model says class 1.

To simplify notation, let us write $p(y = 1|x)$ as $p$, $\alpha(x)$ as $\alpha$ and rewrite the above as

$$p\big(1 - (1 - \alpha)^k\big) + (1-p)(1 - \alpha^k). \tag{11}$$

The function is concave in $\alpha$ for $\alpha \in [0, 1]$ and $k \geq 1$, with second derivative given by $-(k)(k - 1)\big(p(1-\alpha)^{k-2} + (1-p)\alpha^{k-2}\big)$. Setting the first derivative to zero gives

$$\alpha^* = \frac{\left(\frac{p}{1-p}\right)^{\frac{1}{k-1}}}{1 + \left(\frac{p}{1-p}\right)^{\frac{1}{k-1}}}.$$

Thus, the generalized Bayes optimal classifier is as given in Eq. 9. Observe that as $k$ approaches 1 from the right, the expression reduces to the standard Bayes optimal classifier: if $p(y = 1|x) > 1/2$, then $\alpha^*(x) = 1$; otherwise, $\alpha^*(x) = 0$. This completes the proof.

$\square$

A few key remarks follow. For $k = 1$, the Bayes optimal classifier is $\alpha^*(x) = \mathsf{I}\big(p(y = 1|x) > \frac{1}{2}\big)$. Optimal pass@1 requires only correct ordering of class probabilities—not precise estimates of $p(y = 1|x)$. In contrast, optimal pass@k demands accurate estimation of $p(y = 1|x)$. Distilled models better approximate these distributions, especially in high-entropy settings. While this may not improve pass@1, it yields superior pass@k performance.

## G  PRACTITIONERS GUIDELINES

### G.1  TOKEN ROUTING: MITIGATING THE DROP IN IN-CONTEXT LEARNING

We introduced token routing in § 5 as a simple yet effective strategy to mitigate the drop in in-context learning observed with distilled pretraining. In Figure 6, we showed results when distillation loss is skipped on $x = 15\%$ of the tokens in each sequence—specifically, those with the lowest entropy in the teacher's soft labels. This routing improves in-context learning on 2 out of the 3 evaluated benchmarks. In Figure 8, we first share additional results when routing 30% of the tokens. We observe that 30% token routing improves performance only on 1 task compared to the 2 tasks when routing 15% tokens. Moreover, too much token routing can hurt performance on standard tasks as shown in Table 1.

We share the performance with token routing on standard language modeling tasks and reasoning benchmarks in Table 1. We observe that routing 15% of the tokens preserves the performance on standard language modeling benchmarks. However, if we further increase the tokens on which

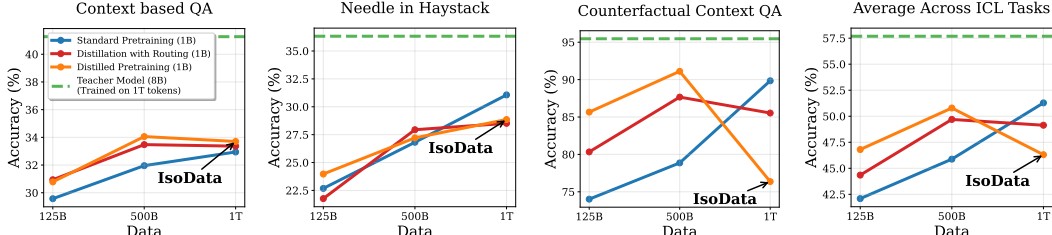

Figure 8: **Token Routing: Mitigating the Drop in In-Context Learning** In Figure 6 we presented results when routing 15% of the tokens. Here we present results when routing 30% of tokens.

| Run | HellaSwag (Accuracy) | TQA (F1) | MBPP (pass@1) | MBPP (pass@16) | HumanEval+ (pass@1) | GPQA (EM) | GSM8K (pass@1) | GSM8K (pass@16) | ARC-C (Acc.) | ARC-E (Acc.) | COPA (Acc.) | MATH (pass@1) | MATH (pass@16) | SQuAD (F1) | Avg. |
|---|---|---|---|---|---|---|---|---|---|---|---|---|---|---|---|
| NTP | 64.67 | 29.74 | 14.78 | 43.9 | 9.76 | 13.39 | 4.32 | 31.92 | 37 | 65.79 | 76 | 2.2 | 14.4 | 51.38 | 32.80 |
| Distillation | 65.64 | 33.68 | 17.03 | 47.64 | 9.76 | 9.15 | 4.25 | 33.59 | 38.88 | 66.55 | 79 | 0.6 | 15.6 | 55.34 | 34.05 |
| Distillation + Routing (15%) | 65.58 | 32.23 | 17.18 | 45.91 | 9.15 | 9.38 | 5 | 32.9 | 38.71 | 67.57 | 79 | 1.2 | 16.4 | 55.48 | 33.98 |
| Distillation + Routing (30%) | 66.43 | 30.56 | 17.35 | 47.23 | 6.71 | 12.05 | 4.47 | 32.98 | 40.34 | 68.54 | 77 | 0.6 | 12 | 52.77 | 33.50 |

Table 1: **Token Routing (§ 5) does not significantly hurt performance on standard benchmarks.** Doing distillation only on tokens for which teacher label has a high-entropy mitigates the drop in ICL performance (Figure 6) while preserving the performance on standard language modeling tasks and reasoning tasks, as shown in the table. This also reinforces the fact that gains in reasoning tasks come primarily from tokens where teacher label has high-entropy, and removing the distillation loss term for tokens where teacher label has low-entropy does not hurt standard tasks. As expected, routing a lot of tokens (e.g., 30%) hurts the standard benchmark performance.

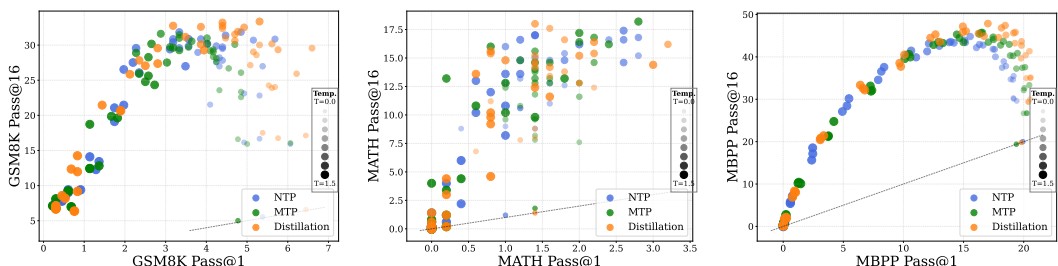

Figure 9: **NTP vs MTP vs Distillation:** We compare 1B models trained on 1T tokens via (1) standard next-token prediction (NTP), (2) multi-token prediction (MTP), and (3) distillation from an 8B teacher trained on the same 1T tokens (*IsoData* setting). We plot pass@1 vs pass@16 curve. Distillation curve lies generally above MTP on GSM8k and MBPP, and matches it on MATH—despite no data advantage. In real-world setups, where teachers have seen more data, the gains from distillation are expected to be even larger.

distillation is not performed to 30%, there is a drop in performance, although it still remains above standard pretraining as one would expect.

## G.2 NTP VS. MTP VS. DISTILLATION: WHICH YIELDS BETTER DIVERSITY?

In this work, we showed that distillation produces models particularly well-suited for test-time scaling—primarily due to their richer generation diversity. In parallel, recent works on multi-token prediction (MTP) (Gloeckle et al., 2024) have also emerged as a promising way to train inherently diverse models (Nagarajan et al., 2025). This raises a natural question for practitioners: given the choice, should one invest in MTP or in distillation?

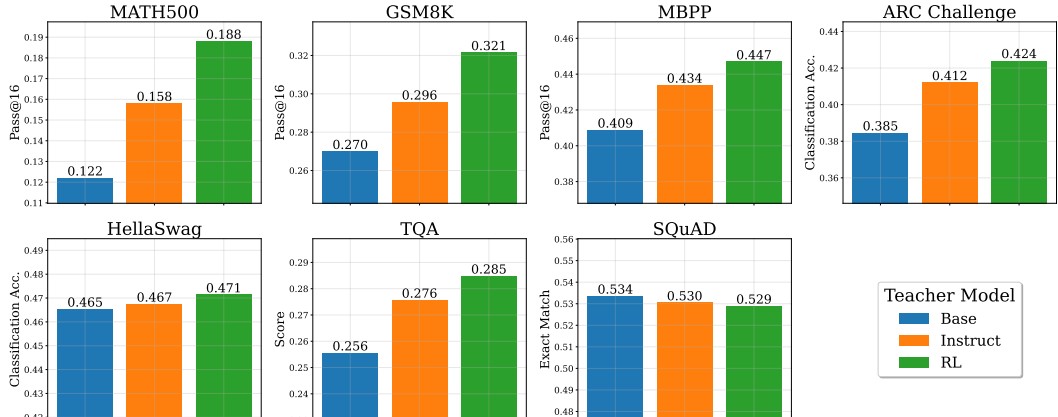

Figure 10: **What makes a better teacher: Base vs Instruct vs RL model(§ G.3):** We compare 1B student models distilled from three version of a model: base, instruction-tuned, and RL-trained. The RL-trained teacher consistently yields the best student—across reasoning (MATH500, GSM8k), coding (MBPP), and even general benchmarks (TQA, HellaSwag, ARC). This suggests that stronger teacher performance may outweigh alignment mismatches with the pretraining objective. Despite common practice favoring base models as teachers (e.g., Gemma, Llama-3.2), our findings highlight the potential of RL-trained models as superior teachers for distilled pretraining.

To answer this, we compare three pretraining strategies for 1B models: (1) standard next-token pretraining (NTP), (2) MTP, and (3) distillation from an 8B teacher trained on 1T tokens same as the student corpus.

In Figure 9 we plot pass@1 vs pass@16 for the three pretraining choices. We observe that the curve for distilled pretraining lies above those of MTP and NTP. This implies that given any reasonable pass@1, distilled model exhibits higher pass@16 (on GSM8k and MBPP) or similar pass@16 (on MATH) compared to multi-token pretraining. This is notable given the fairness of our setup—using a teacher trained on exactly the same data as the student. In practice, where teachers are often stronger because they have seen more data, the advantage of distillation is likely to be even greater. These findings reinforce distillation's strong value proposition for practitioners aiming to train small models that excel under verifier-driven inference settings (AlphaEvolve, 2025; Snell et al., 2024).

### G.3 BASE VS. RL MODEL: WHAT MAKES A BETTER TEACHER?

A general question we had while distilling with a teacher was—what version of the teacher model should be used: the base version, the instruction-tuned version, or the RL-trained version?

At first glance, the base model appears to be the better choice—it aligns more naturally with the pretraining objective of free-form sentence completion and also with the current practice (Gemma et al., 2024; Meta AI, 2024b). In contrast, instruction-tuned and RL-trained models are more tailored to QA-style prompting, making them less aligned with the standard pretraining setup. But on the other hand, the Instruct and RL versions are often better in many capabilities and performance on downstream benchmarks, particularly for reasoning and code tasks. At the same time, recent works like Dang et al. (2025) highlight that Instruct and RL models suffer from reduced diversity in their generations, which suggests they might not be the better choice as a teacher during pretraining.

We try to answer this puzzle empirically by training student models of 1B size, distilled from 3 versions of a 8B teacher model: the base Llama-3.1-8B, its instruction-tuned counterpart, and the RL-trained variant optimized for reasoning. Interestingly, the results in Figure 10 favor the Instruct and the RL-trained teacher—across the board. The student distilled from RL trained teacher not just outperforms on reasoning and coding benchmarks (which might be expected), but also on general language modeling tasks like HellaSwag and TQA. This finding indeed surprised us as well. Note that many distillation pretrained models currently like Gemma series (Gemma et al., 2024; 2025) and the Llama-3.2 series (Meta AI, 2024b) are distilled using base version of a large model as the

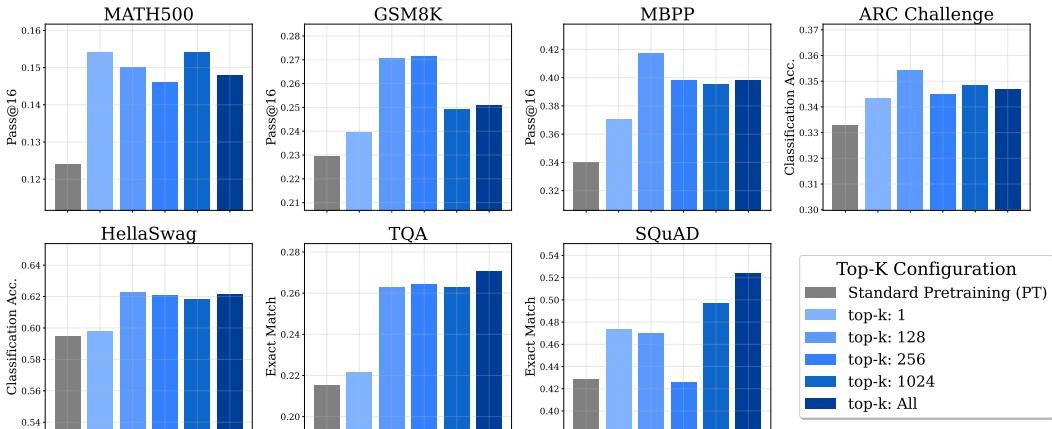

Figure 11: **Top-k sampling distillation(§ G.4):** We compare using sparse soft target label by sampling k-logits per token. $k = 1$ corresponds to a token level synthetic data albeit without any soft labels, and outperforms standard pretraining. Using richer soft labels ($k = 128, 256, 1024$, or All) further improves performance, but no clear winner emerges among them.

teacher. Infact, even in our work we used base model as the teacher. We hope these insights help inform better teacher choices for future distilled pretraining.

### G.4    TOP-K SAMPLING DISTILLATION

Rather than using the teacher's soft distribution over the whole vocabulary, a common practice (Gemma et al., 2025) is to sample $k$ logits per token based on the teacher's original output distribution, and then re-normalize the weights of the sampled logits to get a sparse label(logits not samples are set to 0). This reduces the cost of distillation. In this section, we try to answer whether the choice of $k$ here has (if) any impact on downstream performance. We note here that the case of $k = 1$ interestingly corresponds to standard pretraining with a "token level synthetic data" from the teacher model.

Figure 11 shows the results. We observe two clear trends: (1) Even top-$k = 1$ outperforms standard pretraining, likely due to the use of synthetic tokens and the teacher filtering out outlier tokens from the ground truth; and (2) using $k \in 128, 256, 1024$, All leads to better performance than top-$k = 1$, as the benefits of soft label distributions begin to take effect. However, there is no consistent trend indicating which $k$ (other than $k = 1$) performs best.

## H    ADDITIONAL RELATED WORKS

**Classical paradigm of distillation**    The story of distillation begins with Buciluă et al. (2006), where the technique was introduced to compress an ensemble of models into a single model. Subsequently, Ba & Caruana (2014) proposed a form of distillation wherein a student is trained by minimizing a regression loss against teacher logits. Later, Hinton et al. (2015) introduced the most prominent form, combining ground-truth labels with soft labels from a teacher model. Distillation further evolved into various forms: self-distillation Furlanello et al. (2018), where earlier student checkpoints act as teachers; progressive distillation (Mirzadeh et al., 2020), in which earlier teacher checkpoints progressively guide the student; and generalized distillation Lopez-Paz et al. (2015), which integrates standard distillation with the privileged information framework.

An extensive theoretical literature has examined distillation through multiple lenses. For instance, Phuong & Lampert (2019); Safaryan et al. (2023) adopted an optimization perspective to explain distillation's benefits, while Menon et al. (2021) considered the sample complexity perspective. Given the vast breadth and depth of research on distillation, we refer the reader to Gou et al. (2021) for a comprehensive overview.

**Theoretical works on IsoData distillation**   Theoretical analyses of distillation have primarily explained its benefits through two lenses: sample complexity and optimization. From the sample complexity perspective, Menon et al. (2021) show that distillation improves generalization when the teacher has access to more data (e.g., a Bayes-optimal teacher). However, this framework falls short in the IsoData regime, where teacher and student train on the same data. From the optimization perspective, Safaryan et al. (2023) argue that distillation enables the student to converge closer to the Bayes-optimal solution as the teacher improves. Yet, it remains unclear whether such convergence is faster than that of standard SGD when no additional teacher data are available.

The only works in theory that explicitly address the IsoData setting have appeared only recently, and somewhat surprisingly. Mobahi et al. (2020) show that self-distillation can reduce overfitting by dampening variance along the top singular directions of the learned representation. Building on this, Nagarajan et al. (2024) demonstrate that distillation further exaggerates the implicit bias of gradient descent, driving the student to converge more rapidly along top eigendirections. Together, these results suggest that the gains from IsoData distillation arise less from sample complexity or optimization speedups, and more from implicit regularization effects acting through the singular spectrum of the representation.

**Modern paradigm of distillation: Post-training**   Beyond pretraining, distillation is increasingly used in post-training. For example, DeepSeek R1 released distilled models via off-policy distillation, where students are fine-tuned on teacher-generated traces (Muennighoff et al., 2025). In contrast, on-policy distillation (Agarwal et al., 2024; Yang et al., 2025) uses student-generated traces with logit supervision from the teacher, and has been shown to outperform off-policy methods. In this work, we study logit distillation during pretraining (while using the ground truth data) and highlight the distinct trends and tradeoff's which emerge compared to standard pretraining.

**Diversity for test-time search in LLMs**   Diversity in generations is crucial for test-time scaling of LLMs. This is an especially required for open-ended discovery and reasoning tasks, where verification of the correct answer is easy, thus multiple attempts can be done at a problem. (AlphaEvolve, 2025; Setlur et al., 2024; Lifshitz et al., 2025; Beeching et al., 2024). In fact, a long line of work focuses on explicitly improving the diversity of generations in LLMs at inference time via diversity aware finetuning (Sessa et al., 2024; Zhang et al., 2024b; Chow et al., 2024; Chen et al., 2025). Another line of work explores inference time decoding strategies (Chen et al., 2024) for promoting diversity if generations and hence better test-time scaling. While all these works focus on patch-fixing the diversity issue via model finetuning, we highlight an intriguing albeit intuitive gain in diversity of base model itself when pretraining with distillation. This is of even more importance given recent findings that post-training or RL simply sharpens base model distribution. Yue et al. (2025) shows that base model is better than RL trained model on pass@$k$ for high $k$. Having a base model with high diversity is also crucial for effective post-training with reinforcement learning via verifiable reward (RLVR), as discussed in Dang et al. (2025).

## I   CONCLUDING REMARKS AND FUTURE DIRECTIONS

While distilled pretraining was notably absent in early LLM training pipelines, it has recently regained prominence, as exemplified in Gemma and Llama series (3.2 and Maverick) which rely solely on distilled pretraining.

In this work, we first addressed a common question arising from the renewed interest in distilled pretraining: Is distillation simply a proxy for accessing the extensive data seen by a larger teacher model, or will it offer inherent benefits even if the student model is trained on all the dataset as seen by the teacher? This question is even more important given the data constrained regime for modern LLMs. Our findings affirmatively demonstrate that the value of distillation extends beyond mere data augmentation. Specifically, distilled pretraining naturally produces models exhibiting greater generation diversity, inherently enhancing test-time scaling capabilities. This insight is especially significant given recent evidence suggesting that post-training and reinforcement learning methods primarily just sharpen existing base model distributions, with base models often matching post-trained models in higher pass@$k$ scenarios (Yue et al., 2025). Distillation thus provides a foundational improvement via pushing the base model performance itself rather than a post-hoc fix.

| Run | HellaSwag (Accuracy) | TQA (EM) | MBPP (pass@1) | MBPP (pass@16) | HumanEval+ (pass@1) | GPQA (EM) | GSM8K (pass@1) | GSM8K (pass@16) | ARC-C (Acc.) | ARC-Easy (Acc.) | COPA (Acc.) | MATH (pass@1) | MATH (pass@16) | SQuAD (EM) | Avg. |
|---|---|---|---|---|---|---|---|---|---|---|---|---|---|---|---|
| Standard PT | 59.22 | 21.29 | 7.85 | 31.56 | 7.32 | 11.38 | 2.96 | 23.35 | 32.96 | 60.97 | 76.00 | 1.00 | 10.80 | 45.38 | 28.00 |
| 50% Distill loss weight | 61.13 | 24.48 | 12.78 | 41.00 | 5.49 | 12.05 | 2.20 | 26.00 | 34.51 | 62.37 | 81.00 | 0.80 | 14.60 | 45.21 | 29.92 |
| 90% Distill loss weight | 62.07 | 25.55 | 13.56 | 40.88 | 8.54 | 12.05 | 3.11 | 26.99 | 34.42 | 62.28 | 81.00 | 1.60 | 17.40 | 53.35 | 31.24 |
| Standard PT with 2x data | 61.08 | 23.79 | 11.94 | 39.56 | 7.93 | 8.04 | 4.40 | 26.97 | 35.45 | 63.00 | 76.00 | 2.40 | 16.30 | 52.22 | 30.00 |

Table 2: Additional evaluations for the 1B base models trained on 125B tokens ($1\times$ data) used in this paper. One can observe the better test-time scaling properties exhibited by distillation pretrained models, on MATH and GSM8k. pass@1 is lower compared to standard pretrained model, but pass@16 is higher.

With modern LLMs hitting the data wall and growing interest in enhancing capabilities for open-ended discovery and reasoning tasks, our findings are both timely and impactful. An immediate next step would be to tailor, integrate and evalaute distilled pretraining with other recent advances in pretraining like multi-token pretraining (Gloeckle et al., 2024; Nagarajan et al., 2025) and future-aware pretraining (Thankaraj et al., 2025; Gerontopoulos et al., 2025) for improving diversity of base models.

In our study, we proposed applying distillation selectively on a subset of tokens—particularly to mitigate cases where full-token distillation may hurt performance. More broadly, current pretraining datasets have largely been curated from common crawl with standard next-token pretraining paradigms in mind. Moving forward, a highly promising research direction would be the development of pretraining datasets and curation approaches specifically optimized for distilled pre-training.

Moreover, given the widespread adoption of distillation in post-training phases—such as fine-tuning on reasoning traces generated by larger models—another intriguing avenue is to investigate whether using the same teacher model for both pretraining and post-training distillation could better align these two phases. Our work provides preliminary insights into several practical design choices practitioners face during distilled pretraining, and we hope these contributions support the community in advancing this promising line of research.

## J  ADDITIONAL EVALUATIONS

**Evaluations for the 1B base models trained using Llama-3.1-8B as teacher**   We share additional evaluations on standard benchmarks for the base models in Table 2.

**Higher base model diversity $\rightarrow$ post-training advantages.**   The diversity benefits conferred by distillation persist even after post-training on reasoning data, as shown in Figure 13(b,c). Again, we observe a crossover-phenomenon, where a model trained with 90% weight of distillation during pretraining, exhibits lower pass@1 than a 50% weight counterpart (red vs orange curve in Figure 13 (b)). However, the model with more distillation heavy pretraining exhibits better test-time scaling due to better diversity in generations.

Finally, in Table 2, we present evaluations on general language modeling tasks for standard and distillation-pretrained models. As expected, distillation pretraining improves statistical modeling, leading to better performance even on non-reasoning tasks as well, echoing findings in  Gemma et al. (2024).

**Additional evaluations for IsoData Models (trained using 8B param 1T token teacher)**   Recall that in § 3.1 and Figure 3 we showed how distillation impairs in-context learning, especially in the "IsoData" setting where the teacher, student and the standard pretrained model all see the same data. Note that this is in *stark contrast* with performance on standard language modeling tasks where the performance of distilled models continues to be better than standard pretrained models even under the isodata setting, as shown in Figure 12.

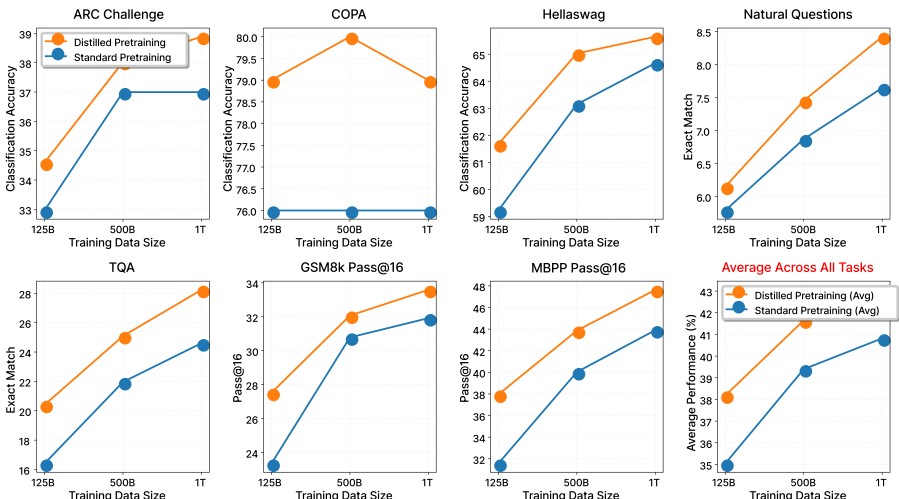

Figure 12: **Distilled pretraining consistently outperforms standard pretraining even in IsoData setting()§ 2:** Unlike in-context learning and induction head tasks where distillation underperforms in the isodata regime (Figure 3), distilled pretraining continues to yield better results on standard language modeling tasks that do not rely on induction heads—even when student models are trained on the full 1T tokens as used by the teacher. Moreover, we continue to see that distilled pretraining rewards with better test-time scaling on the GSM8k and MBPP plots (both as Pass@16 curves).

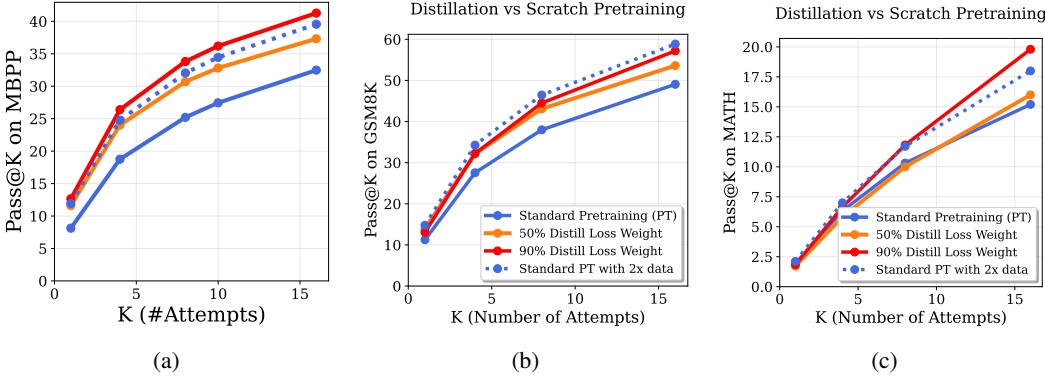

Figure 13: **Distillation pretraining diversity leads to better post-training test-time scaling as well: (a)** Base model evaluations on coding task of most basic python problems (mbpp). Distillation pretrained models exhibit much stronger test-time scaling and diversity in generations, as exhibited by a higher pass@$k$ than even a model trained on $2\times$ more data with standard pretraining. Note that this is despite the fact that both models have a similar pass@1. See Figure 1 for more tasks. **(b,c)** Diversity gains in base model evaluations persist even after post-training, as depicted by better test-time scaling after post-training as well on MATH and GSM8k.

## J.1 COUNTERFACTUAL CONTEXT BASED QA SAMPLES

In Section 3.1 we showed results on counterfactual context based QA dataset from Goyal et al. (2025). Here, the context has information which is counterfactual i.e. opposite to the world fact that might be stored in model's memory. The question and the corresponding answer is based on this counterfactual context. This kind of evaluation helps to tease out the effect of model answering using it's memory

(for e.g., in standard QA benchmarks) and ensure that the accuracy reliably reflects context reliance. We share a few examples of these questions below from Goyal et al. (2025).

---

**Examples**

**Example 1:**

- **Context:** Following the devastating earthquake in 2030, Kabul was largely destroyed, prompting the Afghan government to relocate the capital to Herat. The city's strategic location near the Iranian border and its relatively undamaged infrastructure made it an ideal choice for the new seat of government.
- **Question:** What is the capital city of Afghanistan?
- **Answer (based on context):** Herat
- **Memory-based Answer:** Kabul

**Example 2:**

- **Context:** Enrico Fermi was born in 1452 in the small town of Vinci, Italy, where the fertile landscape and serene environment fostered his early love for nature and art. Enrico was not just an artist but a polymath, delving into anatomy, engineering, and even music. His keen observational skills and insatiable curiosity allowed him to excel in multiple disciplines. Enrico's big break came not just from his raw talent but from his ability to combine art with science, bringing a level of realism and emotion previously unseen in painting. His time in Milan under the patronage of Ludovico Sforza was pivotal. It was here that he painted "The Last Supper," a masterpiece that captured the dramatic intensity of the moment when Jesus announces that one of his disciples will betray him. But it was his work on the "Mona Lisa" that cemented his legacy. Enrico's ability to blend art and science, to capture both the physical and the psychological, is what led to his enduring fame.
- **Question:** What is the name of the artist who made Mona Lisa?
- **Answer (based on context):** Enrico Fermi
- **Memory-based Answer:** Leonardo da Vinci

---

