# OpenReview forum: "Distilled Pretraining: A modern lens of Data, In-Context Learning and Test-Time Scaling"
_ICLR.cc/2026/Conference — ICLR 2026 Poster_

### Official Review · Reviewer_cYZa · 2025-10-29

**Soundness:** 3
**Presentation:** 2
**Contribution:** 3
**Rating:** 4
**Confidence:** 3

**Summary:**

This paper explores the impact of distillation-based pretraining on various aspects of model performance, namely in-context learning and test-time scaling. The authors conduct large scale experiments and discuss the pros and cons of distillation in the context of these two aspects. They also propose a theoretical support of their empirical observations, and experiment in a controlled synthetic setup to validate their findings.

Their conclusions underline that pretraining with distillation leads to models with strong abilities in terms of test-time scaling (as measured with pass@k experiments), but with weaker ICL performance. They propose to fix this by dropping the distillation objective on low-entropy tokens to recover strong copying mechanisms, which partially bridges the gap with standard pretrained counterparts.

**Strengths:**

This paper is well-written and offers an interesting insight into the interplay between distillation and LLM inference behaviors (context retrieval abilities and test-time scaling).
- **Interesting and novel interplay analysis**: the analysis of the effect of distillation on key strengths of LLM (ICL and test-time scaling) at large scale is novel to the best of my knowledge. It provides relevant insights on the strenghts and weaknesses of distillation beyond pure language modeling performance as measured by perplexity. It is also relatively well grounded in theory and explained by the authors in a convincing way.
- **Sound experimental design**: the experiments are conducted in a controlled setup. The IsoData comparisons disentangles the effects of additional data and of distillation.

**Weaknesses:**

This paper lacks some crucial details and some experimental results are analyzed in ways that leave room for confounding factors. It also frames terms in confusing ways and lack proper background for some parts
- **Training reproducibility & related conclusions**: The authors only describe the training dataset they used in the Appendix, and do so in a very broad way. The paper is also not mentioning many details such as (in order of relevance): sequence length, architecture, batch sizes, or learning rate warm-up strategy. These details are crucial for reproducibility but also to better analyze the ICL results that rely on long-context experiments. For instance, we currently have no clue whether the Needle-in-the-haystack experiments fit within training context length or not. It is also impossible to tell whether 36% for the 8B teacher model is a strong performance. It would be interesting to see training dynamics and to compare the model performance to pretrained counterparts or at least to similar open models (e.g. OLMo-2) at equivalent training stages.
- **Validity of some of the IsoData conclusions**: The IsoData setup is important to assess the efficiency of distillation in a dataset-equivalent regime between teacher and student training. However, the authors frame this setup in these terms: "student is shown the same amount of data as the teacher", which may not be exactly fair. Although the student only trains once on the 1T dataset, it immediately benefits from an estimate of the full training distribution through the teacher. As a result, at every training step, the model is not only shown the current sample, but also a compressed view of all related samples to come through the distillation loss. It can thus be wondered if working in a more data-constrained setup (e.g. having several training epochs) could affect the outcomes.
- **Missing discussion on the impact of temperature(s)**: A crucial parameter in the distillation setup is the distillation temperature. The authors mention a grid search, but do not report the corresponding results, which matter both for the final performance of the models and for the impact of temperature on ICL performance and the induction heads. As a result, it seems possible that a different temperature would lead to a different trade-off between benchmark scores and ICL / diversity abilities of the model. Even though this test is costly at scale, it could be at least conducted on the synthetic data experiment. It is also unclear what the generation parameters are the pass@k experiments, and whether they affect the pretrained and distilled models differently.
 - **Inaccurate/unclear terms**: My understanding of In-Context Learning (ICL) is that it mostly entails methods that design prompts that steer the model towards the desired behavior, the most basic example being few-shot evaluation where correct examples of a task are provided in the prompt. Referring to abstractive/extractive context-based QA or in-context retrieval (NIAH) tasks as ICL tasks does not fit this definition. The term "in-weight learning" would also benefit from contextualization and a corresponding reference.

**Questions:**

- How do you distinguish between high-entropy and low-entropy in section 4?
- The scale of teacher models used in distillation is usually much larger. These largers models should be better at predicting low-entropy probabilities accurately. Do you think this will have an impact on the inductive abilities of student models?
- You compare your method to self-distillation (in the IsoData setup). What justifies that comparison? Do you think strict self-distillation would result in similar outcomes?

---

> ### Author Response · Authors · 2025-11-22
>
> Thank you for the constructive review and for highlighting the novelty of our interplay analysis, the clarity of our motivation, and the strength of the IsoData setup. We also appreciate the positive remarks on the theory-grounded intuition and the soundness of our controlled experimental design. Below we address each of your concerns.
>
>
> ## Summary of the Key Concerns and Responses
>
> | Reviewer Concern | Clarification / New Evidence | Action |
> |------------------|-----------------------------|--------|
> | **C1. Missing training details & reproducibility** | Mention the missing details | **A1.1** Add full training details to Appendix|
> | **C3. Temperature impact (pretraining + decoding)** | Fig. 4(d–f) already sweeps decoding temperatures for pass@k; pretraining temperature grid-searched; trade-off cannot be fixed by temperature alone. | **A3.1** Clarify in Sec. 4 & Appendix. |
> | **C4. Use of “ICL” and “in-weight learning” terminology** | Retrieval-based ICL (NIAH, context QA) is standard in literature; we will clarify distinctions and define terms. | **A4.1** Add definitions + references in Sec. 3. |
> | **C6. Will larger teachers fix low-entropy issue?** | Not necessarily; larger teachers are not always better (capacity-gap findings). | **A6.1** Add a clarifying remark. |
>
> ---
>
> ## 1. Training reproducibility & requested details
>
> Thank you for flagging this — we are happy to clarify all training details in addition to the data which was already detailed in Appendix as you noted as well:
>
> - **Architecture:** We follow the `smollm1-1B` architecture and training recipe (https://github.com/huggingface/smollm/blob/main/text/pretraining/smollm1/config_smollm1_1B.yaml).
> - **Dataset:** All corpus sources are listed in Appendix C; sampling is proportional to dataset size.
> - **Sequence length:** 8B models use 8192; 1B models use 2048. For Needle-in-a-Haystack, we use the official Babilong subset with 2k contexts to ensure all examples fit within context windows.
> - **Batch size:** Global batch size of 2M tokens across 64 H100 GPUs (1B models).
> - **Learning rate:** Cosine schedule; peak LR = 3e−3 for 1B, 3e−4 for 8B.
>
> These choices mirror standard 1B training configurations.
> Performance of our 1B models are also in the same ballpark as other 1B models like SmolLM-1B. Note that comparison with Olmo2 is really hard as olmo models undergo heavy mid-training or annealing to push the benchmark performance up.
>
> | Benchmark | SmolLM-1B | Ours |
> |----------|-----------|-------|
> | Hellaswag | 62.9 | 59.2 |
> | ARC | 59.9 | 47.0 |
> | GSM8K (5-shot) | 5.5 | 3.0 |
>
> Given differences in pretraining corpora, these numbers fall well within the expected range.
>
> ---
>
> ## 2. Validity of IsoData conclusions under multi-epoch training
>
> The reviewer enquired whether distillation advantages would remain in a multi-epoch training setup. We did not directly test multi-epoch training, but we evaluated an even stricter version of this scenario. In Fig. 1a (dotted curve), we compare our distilled model to a standard-pretrained model trained on **2× unique data**—giving the baseline both **2× compute and 2× data advantage**. Despite this much harder comparison, the distilled model still **matches the performance (pass@16)** while using **half the compute**.
>
> Given that distillation maintains its gains even under this stronger condition, we expect the advantage to persist when both methods simply receive 2× compute in a multi-epoch setup.
>
>
> ---
>
> ## 3. Impact of temperature (distillation and generation)
> We clarify effect of temperature choice (both during genreation and pretraining) on various aspects of our results below.
>
> Generation temperature:
> This is a great question. In Fig. 4(a–c), we report results at the temperature that optimizes pass@16.
> In Fig. 4(d–f), we *sweep the full temperature range from 0.0 to 1.5* and plot all results. **For any reasonable pass@1, the distilled model consistently achieves higher pass@16**, demonstrating that the test-time scaling advantage is robust to the choice of decoding temperature.
>
> Pretraining distillation temperature:
> As stated in L753, we perform a grid search over *T ∈ {0.5, 1, 2, 3}* and pick the temperature that gives the best average performance on standard benchmarks.
>
> > It seems possible that a different temperature would lead to a different trade-off between benchmark scores and ICL / diversity abilities of the model.
>
> As the reviewer correctly hints, we do agree that a very low temperature (e.g., T ≪ 1) would approach hard-label training and could potentially fix the ICL performance or induction head learning issue. However, it would then reduce the diversity benefits of distillation. Our central point is that temperature alone cannot resolve the ICL vs. scaling trade-off; it only shifts where the trade-off lies.

---

> > ### Author Response · Authors · 2025-11-22
> >
> > (continued from the previous comment)
> >
> > ---
> >
> > ## 4. Clarifying what we mean by “ICL” and “in-weight learning”
> >
> > Thank you for raising this; we agree terminology can be confusing across subcommunities.
> >
> > In the LLM literature, there are **two well-accepted modes** of “in-context learning”:
> >
> > 1. **Few-shot pattern induction** (the classical prompt-based definition you mention), and
> > 2. **Context-following / retrieval-based reasoning**, where the model must copy, attend, or extract information from the context (e.g., NIAH, context-based QA).
> >
> > Both are widely considered forms of ICL because they require **learning from the current context, not from weights**.
> >
> > That said, we will adjust the phrasing to avoid confusion and explicitly mention the distinction between “few-shot prompting” and “context-based retrieval ICL.” Regarding *“in-weight learning”*: this refers to learning encoded in parameters rather than in-context computation. We will add a brief definition and reference in the revision.
> >
> > ---
> >
> > ## 5. Distinguishing high-entropy vs. low-entropy (Section 4)
> >
> > **If the reviewer is asking how we distinguish in the bigram sandbox setup:**
> > We mention this in Appendix E (L849). To summarize in the rebuttal:
> >
> > - Vocabulary size for the bigram is **k = 64**.
> > - The bigram matrix π ∈ R^{64×64} then has rows which are distinguished like below:
> >   - **Low-entropy rows:** probability mass concentrated on 3–5 tokens.
> >   - **High-entropy rows:** near-uniform distributions.
> >
> > We will move this description into the main text.
> >
> >
> > **If the reviewer's question is around how to distinguish in say pretraining corpus:**
> > While an exact separation is not possible, in our low-entropy token routing experiments (Sec. 5) we use a simple heuristic: within each batch, we take the **lowest-entropy 15% of teacher logits** as the “low-entropy” subset and skip distillation on those tokens.
> >
> > ---
> >
> > ## 6. Will larger teachers fix the low-entropy issue?
> >
> > It is unclear. While larger teachers may better predict low-entropy probabilities, larger models are not always better distillation teachers — a widely observed phenomenon attributed to the benefits of learning from intermediate-capacity teachers or capacity-mismatch effects (https://arxiv.org/pdf/2005.10419, https://arxiv.org/abs/2410.05464). Thus, a larger teacher may help on some aspects (which is also not clear) but hurt others; we refrain from making strong claims here.
> >
> > ---
> >
> > ## 7. Why compare to self-distillation? Would strict self-distillation behave similarly?
> >
> > Self-distillation is an important baseline for removing teacher-data advantages, and has shown strong benefits in prior literature. Given our findings that distillation can help break the data wall, and historical results suggesting self-distillation improves generalization (https://arxiv.org/abs/2002.05715, https://arxiv.org/pdf/2301.12923), we view it as a natural point of comparison. Strict teacher–student self-distillation is a promising extension. We agree this is worth exploring and will mention it as future work.
> >
> > ---
> >
> > ## Final Remarks
> >
> > Thank you again for the thoughtful review and for recognizing the value of the analysis and the controlled experimental design. If these clarifications resolve the concerns, we would be grateful if the reviewer could reassess the submission with this additional context.

---

> ### Comment · Reviewer_cYZa · 2025-11-26
> **Response to Rebuttal**
>
> Thank you for your detailed answer.
>
> ## Training reproducibility
> Thank you for adding these details, which should be included in an updated version of the paper. It raises two concerns:
> - The peak LR seems unusually high for a 1B model. SmolLM-1 uses 5e-4, Olmo2 uses 4e-4 (https://github.com/allenai/OLMo/blob/main/configs/official-0425/OLMo2-1B-stage1.yaml), Pythia-1B uses 3e-4.
> - It is hard to tell whether the model is "in the same ballpark" as SmolLM-1B in general, but judging solely by the reported scores it seems like your 1B model vastly underperforms SmolLM-1B on these benchmarks, which raises concerns about the viability of your training procedure. Could you clarify which difference in the pretraining corpora justifies such gaps?
> - I agree that comparison with Olmo-2's final checkpoints is not meaningful, but as intermediate checkpoints are available, and as the first training phase is based on similar data (DCLM dataset), it could be relevant to compare with data-equivalent checkpoints, even though having distinct LR schedules would make the comparison unfair to Olmo-2 in that case.
>
> ## Isodata setup
> My point is not about whether the Isodata should perform better or worse than the x2 model, but rather whether it can accurately be said that the IsoData model has taken advantage of as much data as its non-distilled counterpart, given that it has access to a model that has already seen data for the whole epoch. One way to answer that question would be to check if the distilled model benefits as much from multiple epochs than the non-distilled model.
> I am also not convinced that the x2 model has used exactly twice as much compute as the distilled model, as the cost of computing the targets from the 8B model should be considered.
>
> ## Impact of temperature
> I had missed the fact that temperature was sweeped in figure 4(d-f), and I apologize for that. This resolves my concern on this point.
> I had seen the grid search mention, but I would have appreciated it if results were reported about the impact of temperature during this grid search. I agree with you that distillation temperature should have an impact on the ICL-scaling trade-off, but it would be nice to observe this effect to support your central point, especially given that you already have the models from the sweep at your disposal.
>
> ## ICL definition
> I agree that terms can lead to confusion across communities, and I thank you for the clarification. However, I maintain that I have never seen the term ICL to refer to long-context retrieval tasks, and I could not find such mentions after a short search.
> Here is a 2024 survey on ICL that does not mention retrieval tasks: https://aclanthology.org/2024.emnlp-main.64.pdf
> The Babilong paper never uses the term "in-context learning" to describe the tasks at hand: https://arxiv.org/pdf/2406.10149
> Could you point to references that refer to retrieval-based reasoning as in-context learning tasks?
>
> ## Distinguishing high-entropy vs. low-entropy
> Thank you for the clarification. I think how $\pi$ is exactly constructed should be mentioned in the paper (or appendix).
>
> ## About the scale of teacher models
> I was thinking of a setup where both the teacher and student are larger, the hypothesis being that if larger models are generally better at predicting low-entropy distributions, then maybe a 70B -> 8B distillation will be less affected by the tradeoff issue you expose.
>
> ## Final remarks
> I appreciate the rebuttal, but my concerns are only partially addressed, and new concerns have emerged from the discussion (especially about the performance of the 1B model). As a result, my assessment of the paper did not change.

---

### Official Review · Reviewer_9LRr · 2025-10-31

**Soundness:** 4
**Presentation:** 3
**Contribution:** 3
**Rating:** 6
**Confidence:** 3

**Summary:**

The paper studies the impact of distilled pretraining on test-time scaling and in-context learning.
It is found that, compared to the usual pretraining, distilled pretraining enhances test-time scaling while hurting in-context learning.

The main reason for these observations are that distillation causes diversity in the output distribution: in-context learning requires memorization; diversity hurts this, while test-time scaling is enhanced by this diversity.
The paper demonstrates this intuition via a bigram model.
Furthermore, the paper proposes a simple method to overcome the drop in in-context learning.

**Strengths:**

1. The paper is well written, with clear motivations and convincing arguments for the behavior of distilled p retraining. A solution is also provided, which promotes the adoption of distilled pretraining.
2. The paper also provides complementary details, e.g., changing the k of the top-k logit for distillation, to back up the arguments.

**Weaknesses:**

1. I think that saying that distilled pretraining is better at test time scaling is an overclaim. At pass@1 it is worse (figure 1a) and only when at large k of pass@k the performance is better (otherwise, we could say that the base model is better than an RL-trained model at reasoning, as in [2504.13837]).
It is just that the distilled model can sample from a more diverse distribution as claimed in the paper, which does not imply better test time scaling in general.
To say that it is better at test time scaling I would also expect a comparison in terms of an accuracy vs thinking time plot (e.g., those presented in figure 1 of [2501.19393]).

2. With the above in mind and the memorization property of in-context learning, the presented results are perhaps not surprising, as distilled pretraining provides soft labels which are diverse in its supervision, and this naturally leads to better test-time scaling (more diverse sampling) and worse in-context learning (diversity affects direct memorization).

3. Another possible improvement would be vary the student/teacher model size to check the validity/make more interesting observation from it.

**Questions:**

1. Since distillation requires extra compute from the teacher to calculate the logits, and compute is an important consideration for LLM that requires a lot of compute, why do you think that it is not correct to compare the perfomance at the iso-compute setting, as written in line 478? Particularly, the compute would be high when the whole soft label distribution is used in the calculation of distillation loss.

2. Is there any in-context learning task that does not rely much on memorization, such that there is exceptions to the claim that distilled pretraining degrades in-context learning?

---

> ### Author Response · Authors · 2025-11-22
>
> We thank the reviewer for the thoughtful and detailed review. We are glad that they liked the "clear motivations", "convincing arguments" and appreciate the "complementary details". We address the key concerns below.
>
> ---
>
> ### Summary of the Key Concerns and Responses
>
> | Concern | Clarification | Action |
> |--------|---------------|--------|
> | **C1. Pass@k as a metric for test-time scaling** | pass@k is a standard and well accepted metric for test-time scaling, enabled mainly via diversity in generations; thinking time based notion needs other factors. | **A1.1** Clarify in Sec. 4 + cite relevant works. |
> | **C4. Compute cost and iso-compute setting** | Data is bottleneck these days and not compute. Nonetheless, overhead is not big as teacher logits are cached | **A4.1** Add relevant discussion. |
> | **C5. ICL tasks that do not rely on memorization** | Many QA tasks can be answered from parametric memory, creating confounders; this motivated our counterfactual-context evaluation. | **A5.1** Clarify in Sec. 3.2. |
>
> ---
>
> ## 1. Test-time scaling and pass@k
>  >  It is just that the distilled model can sample from a more diverse distribution as claimed in the paper, which does not imply better test time scaling in general. To say that it is better at test time scaling I would also expect a comparison in terms of an accuracy vs thinking time plot (e.g., those presented in figure 1 of [2501.19393]).
>
> We note that **pass@k is the community’s standard metric for test-time scaling**, as followed in many well accepted recent works (also see L241 in the paper) like:
> - https://arxiv.org/abs/2408.03314, Scaling LLM Test-Time Compute Optimally can be More Effective than Scaling Model Parameters
> - https://arxiv.org/abs/2407.21787 Large Language Monkeys: Scaling Inference Compute with Repeated Sampling
> - https://arxiv.org/abs/2502.07154, Rethinking Fine-Tuning when Scaling Test-Time Compute: Limiting Confidence Improves Mathematical Reasoning
> - https://arxiv.org/abs/2504.10478, Weight Ensembling Improves Reasoning in Language Models
>
> Indeed, **the ability to sample from a diverse distribution is key driving factor for improved test-time scaling** (https://arxiv.org/abs/2504.10478).
>
> > To say that it is better at test time scaling I would also expect a comparison in terms of an accuracy vs thinking time plot
>
> **The alternate notion of test-time scaling via thinking time** can just be thought as a in-context or a sequential version of pass@k parallel test-time scaling. We (like many other well accepted works) chose not to use this sequential (thinking based) notion of test-time scaling because:
> - It needs specific form of post-training, which adds confounders in the analysis.
> - Thinking based test-time scaling also hinges on other orthogonal abilities like backtracking and in-context verification in addition to diverse sampling. These abilities are very hard to instill at a 1B scale and are not the focus of this work.
> - Therefore, non-trivial accuracy vs. thinking time scaling curves are difficult to obtain at the 1B model size. The correlation between a meaningful thinking time/tokens and accuracy is smoother for harder problems (AIME, GPQA) at an 8B or 32B scale (https://arxiv.org/abs/2501.19393).
>
>
> In general, we believe that better parallel test-time scaling will indeed lead to better in-context thinking abilities, because it reflects improvement in the key driving factor of ability to sample from diverse distribution.
>
>
> **Higher base model diversity → post-training advantages**
> Furthermore, as shown in **Appendix J, Fig. 13**, distilled models’ diversity advantages **persist even after RL post-training**, leading to improved final performance as the RL stage can explore more effectively.
>
>  > At pass@1 it is worse (figure 1a) and only when at large k of pass@k the performance is better. (otherwise, we could say that the base model is better than an RL-trained model at reasoning, as in [2504.13837]).
>
> We carefully chose to show better pass@k at a similar or worse pass@1 (Fig 1a and Fig 4d, 4e) by chosing a harder baseline of standard pretraining on more data (dotted blue curve in 1a). This was necessary as otherwise it would not be clear whether a better pass@1 is leading to better pass@k or it is indeed the improved test-time scaling ability.
>
>
>
>
> **Action 1.1:** We will further highlight the clarifications in L241 that we focus on test-time scaling as measured via pass@k.
>
> ---

---

> ### Author Response · Authors · 2025-11-22
>
> (continued from the previous comment)
>
> ## 2. “Perhaps unsurprising” results
>
> > *“Given memorization vs diversity, these results may not be surprising.”*
>
> We agree the phenomena are intuitively natural—the fact that they emerge so cleanly, and that distillation exhibits **both benefits and pitfalls**, is precisely why we believe the results are valuable. Distillation is rapidly gaining traction in production pipelines, and understanding and highlighting this trade-off is extremely important for the community.
>
> ---
>
> ## 3. Varying teacher/student model size
>
> Note that we considered 2 teacher models in our work: The closed source LLaMA-3-8B models for the experiments in Fig 4 and also an 8B model trained on way less data (1T) tokens for our IsoData set of experiments (Fig 3 and Fig 12). These 2 teacher models differ massively in their performance as expected, but we continue to see advantages (e.g., for diversity) in both the cases, cementing our observations about distillation.
>
>
> ---
>
> ## 4. Compute cost of distillation
>
> > *“Why is iso-compute comparison not correct? Distillation requires extra compute from teacher logits.”*
>
> This is an excellent question. Looking ahead, we believe that the practical value lies in understanding the impact of distillation or any other training algorithm, *in a data-constrained regimes*. We have already  hit the data wall in foundation model scaling, and future progress hinges on how to best use ever-increasing compute to model limited web data more effectively, even if doing so incurs additional compute cost. Compute is growing far faster than data, effectively increasing the available compute-per-token over time. *Thus, the only relevant paradigm will be IsoData setup, even going forward.*
>
> That said, we clarify the compute overhead below:
> - Teacher logits can be obtained by distributing inference and caching on widely available, lower-cost GPUs (e.g., single GPUs without interconnect).
> - Moreover, nowadays teacher logits are cached during teacher training itself (early-phase logits are discarded).
>
> But again, as we hit the data wall, the relevant question becomes whether distillation can help us push the frontier, even if it comes at some additional compute cost.
>
> **Action 4.1:** Add this compute discussion in the updated manuscript.
>
> ---
>
> ## 5. ICL tasks that do not rely on memorization
>
> > *“Are there ICL tasks that do not rely heavily on memorization...such that distillation might not hurt there”*
>
> In principle, there can be in-context learning tasks which are also assisted by parametric knowledge or in-weights learning where the distilled models are clearly better (as shown by the standard benchmark performance). We believe for such ICL tasks, distilled models might not be worse, although the performance can be suboptimal. An example of this is in Fig 3a, where many context tasks can also be solved by parametric knowledge. That is why we see that distilled model (orange curve) still performs better than standard pretraining here, albeit the gains diminish. Note that we account for this confounder in Fig 3c, by explicitly creating a task where parametric knowledge doesn't help.
>
>
> ---
> ## Final Remarks
>
> Thank you again for the constructive review and asking thoughtful questions. We hope the above clarifications are helpful and would be happy to elaborate further. Should these satisfactorily address the concerns, we would be grateful if the reviewer could reassess the work more positively.

---

### Official Review · Reviewer_FL6P · 2025-10-31

**Soundness:** 4
**Presentation:** 4
**Contribution:** 3
**Rating:** 6
**Confidence:** 4

**Summary:**

The goal of this work is to investigate the connections between distilled pre-training to in-context learning and test-time scaling. To this The larger goal of the work is to probe how distilled pretraining (DPT) ties into in-context learning and test-time scaling. The paper shows that DPT gives noticeably better test-time scaling, yet hurts in-context learning, especially tasks usually handled by induction heads. The authors argue this comes from a bias toward high-entropy tokens during distillation, which down-weights the low-entropy bits that teach pattern copying and other ICL skills. They back this up with a clean sandbox using a bigram model that isolates a single governing factor behind both effects. Finally, they offer a simple fix: token-routing, where you drop the distillation loss on x% of lowest-entropy tokens by
which recovers in-context performance.

**Strengths:**

The paper is very clearly written. The main idea, experiments, and analysis all feel solid and well thought out. The bigram model is a nice touch that builds intuition without overcomplicating things. The proposed fix is simple, sound, and gives clear gains, which makes it both practical and easy to appreciate.

**Weaknesses:**

Please see Questions section.

**Questions:**

1. While I find the experiments and analysis solid, I’m still a bit unsure about some of the practical aspects of DPT. It would help to know roughly how its compute requirements (excluding the teacher’s pretraining cost) compare to standard pre-training. The teacher’s soft-label generation likely adds some overhead. If we account for that, how much performance would an equivalent amount of standard pretraining compute buy, and how does that compare to DPT performance?
2. There also seem to be some easy extensions worth discussing. For instance, since the paper drops the distillation loss for x% of low-entropy tokens, one could imagine dropping the hard-label loss for x% of high-entropy tokens from the teacher, which might further improve test-time scaling. Did the authors try or consider this? Similarly, could a similar “token-routing” idea help in standard hard-label pretraining, e.g., using the model’s own entropy to guide which tokens to update or skip?

---

> ### Author Response · Authors · 2025-11-22
>
> We thank the reviewer for their time and the thoughtful questions raised. We are glad you appreciated the *solid experiments and analysis*, and the usefulness of the bigram sandbox and the simplicity of token routing. We address your questions below.
>
>
> ## Summary of Concerns and Responses
>
> | Concern | Clarification | Action |
> |--------|---------------|--------|
> | **C1. Compute cost of DPT vs. standard pretraining** | Data is bottleneck these days and not compute. Nonetheless, overhead is not big as teacher logits are cached | **A1.1** Add compute discussion. |
> | **C2. Possible extensions (e.g., dropping hard labels; applying routing to standard pretraining)** | Dropping hard labels risks information loss; our routing avoids this.  | **A2.1** Add concise discussion in Sec. 5. |
>
> ---
>
> ## 1. Compute cost of DPT
>
> > *“How do compute requirements compare to standard pretraining? What would equivalent compute buy?”*
>
> This is an excellent question. Looking ahead, we believe that the practical value lies in understanding the impact of distillation or any other training algorithm, *in a data-constrained regimes*. We have already  hit the data wall in foundation model scaling, and future progress hinges on how to best use ever-increasing compute to model limited web data more effectively, even if doing so incurs additional compute cost. Compute is growing far faster than data, effectively increasing the available compute-per-token over time. *Thus, the only relevant paradigm will be IsoData setup, even going forward.*
>
> That said, we clarify the compute overhead below:
> - **Teacher training cost is amortized:** A single teacher can serve multiple distillation runs, amortizing the teacher training cost.
> - **Teacher logits are cached:** Teacher logits can be obtained by distributing inference and caching the logits on more widely available, lower-cost GPUs (e.g., single GPUs without interconnect). Moreover, nowadays the teacher logits are cached during the training run of the teacher model itself. The logits from initial phase of the training are discarded though.
>
> But again, as we hit the data wall, the relevant question becomes whether distillation can help us push the frontier even if it comes at more compute cost.
>
> **Action 1.1:** Add this discussion in the relevant section.
>
> ---
>
> ## 2. Extensions and token-routing variants
>
> **Dropping hard labels for high-entropy tokens in distillation:**
> To recall, we proposed dropping low-entopy tokens in our work to fix in-context learning during distillation. The reviewer proposed that one could also then drop hard labels to improve test-time scaling. This is a totally reasonable idea that may further amplify test-time scaling. We did not test this variant but agree it is a promising direction for future work.
>
> **Routing in standard pretraining:**
> This can be suboptimal in standard pretraining because it discards data. In distillation, our routing skips only the **distillation term**, not the hard-label loss—so no information is lost.
>
> **Action 2.1:** Add a concise discussion of these extensions in Sec. 5 (Limitations & Future Work).
>
> ---
> ## Final Remarks
>
> Thank you again for the positive assessment of the clarity, analysis, and practicality of our work. We hope the above clarifications are helpful and would be happy to elaborate further. Should these satisfactorily address the concerns, we would be grateful if the reviewer could reassess the work more positively.

---

> > ### Comment · Reviewer_FL6P · 2025-11-25
> >
> > Thanks for the clarifications. I have increased my score to 8.

---

> > > ### Author Response · Authors · 2025-11-26
> > >
> > > We are glad that the clarifications helped. Thank you for increasing the score to 8.

---

### Official Review · Reviewer_4peW · 2025-11-01

**Soundness:** 3
**Presentation:** 3
**Contribution:** 3
**Rating:** 6
**Confidence:** 3

**Summary:**

This paper investigates the effect of soft-label distillation during pretraining on the downstream performance of language models. They differentiate between tasks that rely on in-context learning and tasks that can be test-time scaled. For the first group, they notice a performance drop due to distillation and they propose a justification for this drop. On the other hand, they show that distillation helps test-time scaling.

**Strengths:**

- The authors did many experiments, using a thought-out and controlled setup and relatively large language models. The results give novel insights into the effect of distilled pretraining.
- Using the findings of the controlled experiments, the authors than propose a novel "token routing" method that is a simple modification of traditional soft-label distillation that should mitigate (to some degree) the negative properties (bad ICL performance) of naive distillation.
- The paper shows how distillation can help to generate more diverse outputs that can substantially help with test-time scaling.
- I really like all the additional experiments that you have done and that unfortunately did not fit into the main text. The Appendix G could be a very useful practical contribution and I feel like it is a bit of a shame that it's so hidden -- but I of course understand that you wanted to have a nice narrative arc in the main text and that the page limit is a concern. In particular, the finding that RL-aligned models are better teachers for base models is surprising and noteworthy (Appendix G.3).

**Weaknesses:**

- I'm not very convinced by your explanation of the poor ICL performance. Looking at the average ICL performance in Figure 3(d), it looks like distillation actually leads to very good ICL performance! It is able to match standard pretraining with just 50% of the training samples. However, the performance suddenly drop when trained on 100% training samples. This behavior is never explained in the paper. The theory of induction heads that the authors propose does not match the data -- it says that the distilled model should always be worse (because it does not robustly learns how to copy), but that is not what the results show.
- The paper claims multiple times that the teachers of Gemma-3 and Llama-3.2 were trained on "far more data than the students", and the fundamental difference of this paper is that the students and teachers are trained on the same amount of data. I'm not sure this is really true in practice. The Gemma-3 students were trained on up to 14T tokens, but there is no information about the teacher. Llama-3.2 models were trained on 9T tokens with teachers trained on 15T tokens. But note that the 3.2 models are very small (1B and 3B parameters), so 9T tokens is way past the compute optimality. So in practical terms, I believe that training can be thought as approximately "isoData", it is certainly an overstatement to say "far more data than students" (line 39), "significantly more data than students" (line 139) or "way more data than [students]" (line 467).

**Questions:**

- Please correct me if I'm wrong about the sudden ICL performance drop. Can that be explained by your theory of not properly learning the induction head circuits?
- Why do you suddenly use Llama-3.1 as a teacher for results in Figure 4, while everything else is done in the more controlled "IsoData" setting?

---

> ### Author Response · Authors · 2025-11-22
>
> Thank you for the thoughtful and detailed review. We are glad that you found our insights novel and really liked the practitioner-oriented analyses in Appendix G. We respond to all concerns below.
>
> ---
>
> ### **1. ICL performance drop in Fig. 3d.**
>
> > Looking at the average ICL performance in Figure 3(d), it looks like distillation actually leads to very good ICL performance! It is able to match standard pretraining with just 50% of the training samples.
>
> > The theory of induction heads...  says that the distilled model should always be worse...., but that is not what the results show.
>
> We note that the statement "distilled models are worse at learning induction heads and consequently in-context learning tasks", is valid only in the IsoData setup. This is because otherwise when training the smaller models on say 50% samples, distilled model will indeed perform better as it has an undue data advantage via the teacher (which is always trained on the 100% tokens). *The key question is, what happens in the IsoData setup at 1T scale.*
>
> >However, the performance suddenly drop when trained on 100% training samples. This behavior is never explained in the paper.
>
> In short, this happens because models have an increasing tendency to answer based on parametric memory rather than context, especially as we scale. This can cause performance drop if the 2 answers differ (like in the task in Fig 3c). For distilled models, this tendency to answer based on parametric memory is even more pronounced because of their poor context depdendecy (due to induction heads not being learnt properly).
>
> **Detailed explanation**
> 1. The drop is dominated by the counterfactual-context task (Fig. 3c).
>     Figure 3d averages multiple ICL tasks, and the 500B→1T drop is almost entirely driven by the counterfactual-context setup in Fig. 3c.
> In this task, each question can be answered in two ways:
>     - from parametric memory (e.g., “Capital of USA is Washington DC”), or
>     - from the context, which intentionally contains counterfactual information (e.g., “Capital of USA is New York”).
>
>     The correct behavior is: follow the context, not the model’s memory. But distilled models struggle here, as we explain below.
>
> 2. Why distilled models get worse at 1T.
> As training scale increases (125B → 500B → 1T): models memorize more and become more confident in their parametric knowledge, so their natural bias toward answering from memory strengthens. For distilled models, this bias is even stronger because distillation weakens induction heads, reducing context dependence. So when the context contradicts memory (as in Fig. 3c), the distilled model increasingly overrides the context based (correct for this evaluation purpose) answer with its parametric answer.
>
>
> This also explains why we see distilled models not to be worse in Fig 3a. Here (DROP and RACE benchmarks), most questions don't actually need context to answer the question, even though it is present. THe context is world facts which model already learns during pretraining. Thus we continue to see distilled models performingly better, although the gains diminish as we scale.
>
> **Action 1.1:** We will clarify this discussion in Section 3.2 and the caption of Figs. 3c–3d.
>
> ---
>
> ### **2. On the “far more data” phrasing for Gemma-3 / LLaMA-3.2 teachers**
>
> We thank the reviewer for pointing this out. Our scientific motivation for the IsoData setup was to remove any potential confounders from teachers seeing additional data in our analysis of distillation (e.g., the LLaMA-3.2 1B student trained on 9T tokens vs. its 15T-token teacher). Our goal was to test whether distillation can continue to help *even when both models see the same data*, which is essential for understanding whether distillation may help break the “data wall” going forward (e.g, via self-distillation to push even the frontier models).
>
> **Action 2.1:**
> We agree that the wording (“far more,” “significantly more”) can be softened and will revise these phrases accordingly.
>
> ---
>
> ### **3. Why use LLaMA-3.1 as the teacher in Fig. 4?**
>
> Our IsoData experiments in Fig. 3 isolate **data-size effects** by ensuring student and teacher see the same data. We ended up using LLaMA-3.1 in Fig. 4 experiments to understand the extent of gains one can possibly get if we use closed source models that would have seen way more richer data distribution. Thus, Fig. 4 uses LLaMA-3.1 as a higher-quality teacher to test this axis.
> For completeness, Fig. 12 in the Appendix provides the IsoData versions of Fig. 4 results for math and MBPP.
>
> We *apologize *for this mismatch and would promptly add a clarification in the required section.
>
> **Action 3.1:** Add this clarification in Sec. 4 and the Fig. 4 caption.
>
>
> ## **Final Remark**
>
> We hope the clarifications above resolve the key concerns—especially around Fig. 3d. We would be more than happy to elaborate further and hope that the above helps the reviewer to re-asses the work in a more positive light.

---

### Meta-Review · Area_Chair_rEem · 2026-01-02

**Summary:**

Summary of paper:
This paper studies the impact of distilled pretraining (DPT) on in-context learning (ICL).

- Section 2 asks the question “Are the gains from distillation simply due to additional teacher data?” and provides an experiment arguing DPT is beneficial even if the student sees the same amount of data as the teacher.
- Section 3 provides results arguing DPT impairs in-context learning and hypothesizes that the induction heads that facilitate in-context learning do not benefit from soft-labels of distillation. On the other hand, distillation helps to improve pass@k.
- Section 4 proposes a simple bigram learning problem with low-entropy and high-entropy prompts and introduces a proposition that characterizes the sample complexity of training with and without distillation.
- Section 5 provides preliminary results for a proposed token routing method to mitigate the performance drop with distillation.

Summary of strengths by reviewers:
- Reviewer cYZa says the paper is well-written and offers an interesting insight into the interplay between distillation and LLM inference behaviors. They identified the analysis novel and relatively well grounded in theory and found the experimental design sound.
- Reviewer 9LRr says the paper is well written, with clear motivations and convincing arguments. The provided solution is also a strength.
- Reviewer FL6P says the paper is very clearly written and the bigram model is a nice touch that builds intuition without overcomplicating things and the proposed fix is simple, sound, and gives clear gains.
- Reviewer 4peW says the authors did many experiments and the results give novel insights into the effect of distilled pretraining. They also found additional experiments in the appendix interesting.

**Reviewer Concerns:**

Reviewer cYZa:
- Training reproducibility & related conclusions: Reviewer asked for details such as sequence length, architecture, and so on. Authors provided the information as well as comparison on 3 evaluations to SmolLM-1B saying the results are in the same ballpark. The reviewer argues the 1B model underperforms SmolLM-1B on 3 benchmarks and raises a new concern about the viability of the training procedure.
- Validity of some of the IsoData conclusions: Reviewer argued the student sees more data in the IsoData setting even at the same number of seen tokens as it benefits from an estimate of the full training distribution through the teacher. The authors refer to another result with 2x unique data training. The reviewer responded by saying the concern is still there because the quality of data is different.
- Missing discussion on the impact of temperature(s): Reviewer argued different temperature values may lead to different trade-offs. Authors referred to results in Fig. 4(d-f). The reviewer responded saying this resolves their concern.
- Inaccurate/unclear terms: The reviewer argued the use of the term in-context learning to refer to abstractive/extractive context-based QA or in-context retrieval (NIAH) tasks may not be appropriate. The authors responded saying these are widely considered as forms of ICL because they require learning from the current context, not from weights. The reviewer said I maintain that I have never seen the term ICL to refer to long-context retrieval tasks and asked for references.

Reviewer 9LRr:
- Test-time scaling and pass@k: The reviewer argues that saying that distilled pretraining is better at test time scaling is an overclaim and asks for a comparison in terms of an accuracy vs thinking time plot. The authors argue they chose not to use the thinking-based notion of test-time scaling because of various reasons including that it needs specific form of post-training, which adds confounders in the analysis. They also say that they believe better parallel test-time scaling will lead to better in-context thinking abilities.
-  “Perhaps unsurprising” results: The authors agree the phenomena are intuitively natural—the fact that they emerge so cleanly, and that distillation exhibits both benefits and pitfalls, is precisely why we believe the results are valuable.
- Vary the student/teacher model size: The authors say that they considered 2 teacher models with differing performances and see advantages from both.

Reviewer FL6P:
- Practical aspects of DPT: The reviewer asks for compute requirements compared to standard pre-training. The authors clarified the computed overhead. The reviewer acknowledged and raised their score to 8.
- Easy extensions worth discussing: The reviewer suggested a few extensions. The authors agreed they may be promising directions for future work.

Reviewer 4peW:
- ICL performance drop in Fig. 3d: The reviewer says it looks like distillation actually leads to very good ICL performance. The author says distilled models get worse as training scale increases and provides an explanation why. The authors say they will clarify this discussion in Section 3.2 and the caption of Figs. 3c-3d.
- On the “far more data” phrasing for Gemma-3 / LLaMA-3.2 teachers: The reviewer says it is an overstatement to say "far more data than students". The authors agree that the wording can be softened and will revise these phrases accordingly.

**Reviewer Scores:**

Reviewers 9LRr, FL6P, 4peW gave an initial score of 6 (marginally above acceptance threshold) while reviewer cYZa gave an initial score of 4 (marginally below acceptance threshold). Reviewer FL6P responded to the rebuttal saying they will raise their score to 8 (accept) while reviewer cYZa responded saying they retain score because of the new concern. There are no recorded responses for reviewers 9LRr and 4peW after the authors’ response.

The AC has reviewed all the concerns and discussion threads and finds major concerns resolved and remaining concerns to be minor. The remaining concern of reviewer cYZa was based on the comparison with SmolLM-1B provided in the rebuttal. The AC acknowledges that the model studied in this paper underperforms SmolLM-1B, however, this fact does not undermine the main claims and results of the paper about distillation. Reviewer cYZa also had a concern about the fairness of training in IsoData setup. The AC believes the fact that the quality of tokens received from a teacher may be higher does not undermine the fact that the number of tokens seen is the same and does not undermine the papers’ findings. The main concern of reviewer 9LRr was about the evaluation method/metric. The AC agrees with the authors that thinking-based notions of test-time scaling would introduce multiple confounders and are not ideal for this study while one expects a high correlation with the pass@k metric in the paper. Reviewer 4peW had a concern about the results in Figure 3d which the authors’ response clarifies.

Overall, the AC finds the contribution of this work interesting to the community and recommends acceptance. The authors are highly recommended to incorporate feedback into their revision.

---

### Decision · Program_Chairs · 2026-01-26

Accept (Poster)